# Metabolic cross-feeding in imbalanced diets allows gut microbes to improve reproduction and alter host behaviour

Sílvia F. Henriques [1], Darshan B. Dhakan[1,3], Lúcia Serra[1,3], Ana Patrícia Francisco[1], Zita Carvalho-Santos [1], Célia Baltazar [1], Ana Paula Elias [1], Margarida Anjos[1], Tong Zhang [2], Oliver D. K. Maddocks [2] & Carlos Ribeiro [1✉]

The impact of commensal bacteria on the host arises from complex microbial-diet-host interactions. Mapping metabolic interactions in gut microbial communities is therefore key to understand how the microbiome influences the host. Here we use an interdisciplinary approach including isotope-resolved metabolomics to show that in *Drosophila melanogaster*, *Acetobacter pomorum* (*Ap*) and *Lactobacillus plantarum* (*Lp*) a syntrophic relationship is established to overcome detrimental host diets and identify *Ap* as the bacterium altering the host's feeding decisions. Specifically, we show that *Ap* uses the lactate produced by *Lp* to supply amino acids that are essential to *Lp*, allowing it to grow in imbalanced diets. Lactate is also necessary and sufficient for *Ap* to alter the fly's protein appetite. Our data show that gut bacterial communities use metabolic interactions to become resilient to detrimental host diets. These interactions also ensure the constant flow of metabolites used by the microbiome to alter reproduction and host behaviour.

[1] Behavior and Metabolism Laboratory, Champalimaud Research, Champalimaud Centre for the Unknown, Lisbon 1400-038, Portugal. [2] University of Glasgow Institute of Cancer Sciences, Switchback Road, Glasgow G61 1QH, UK. [3]These authors contributed equally: Darshan B. Dhakan, Lúcia Serra. ✉email: carlos.ribeiro@neuro.fchampalimaud.org

In all organisms, including humans, diet is a critical determinant of health and well-being[1]. As the building blocks of proteins, dietary amino acids (AAs) play a pivotal role in determining the fitness of all animals. Because they cannot be efficiently synthesised, essential amino acids (eAAs) need to be acquired through the diet[2,3]. Furthermore, universally, the over-ingestion of dietary AAs shortens lifespan and negatively impacts healthspan[4–6]. Given the importance of a balanced dietary intake of AAs, organisms are able to direct their feeding choices to homeostatically compensate both for the lack and over-ingestion of AAs[7–13]. In *Drosophila melanogaster* females, both AA deprivation and mating induce changes in specific neuronal circuits which modulate food choice, leading to a drastic increase in protein appetite[9,14]. Remarkably, the removal of any of the 10 eAAs from the fly diet is sufficient to induce this strong protein appetite[8]. While much is known about the physiological and neuronal processes underlying bulk food intake, less is known about how nutrient specific appetites are controlled. Given the importance of a balanced diet and especially of a balanced intake of dietary AAs for animal fitness it is key that we advance our understanding about the factors controlling nutritional choices.

The gut microbiome has emerged as an important modulator of host physiology and behaviour[15–20]. As such, gut bacteria have also been shown to influence feeding behaviour, food choice, and reproduction[8,21,22]. A key challenge of current microbiome research is to identify the mechanisms underlying the effect of the microbiome on the host. While in specific cases single microbes can be identified as the sole drivers affecting the host[23–27], in the majority of cases it is clear that gut microbes act on the host as a community rather than as isolated biotic factors[28–30]. This is likely also the case for the impact of the gut microbiome on the brain. In mice, for example, *Akkermansia* and *Parabacteroides* bacteria have been suggested to act together to mediate the beneficial effects of the ketogenic diet in seizure prevention[29]. Conceptually, two main mechanisms can explain why only a community of microorganisms can affect the physiology and behaviour of the host: (1) the need for the tandem catalytic activity of enzymes from different microbiota for the production of metabolites acting on the host[31,32] and/or (2) the need for an obligatory mutualistic metabolic relation (syntrophy) to sustain the growth of specific microorganisms so that they can promote the observed physiologic effect in the host[33]. To identify the mechanisms by which bacterial communities act on the host, we will need to map out the relevant interactions among gut microbes influencing the host and identify the molecular and metabolic mechanisms by which they do so. This remains a daunting task given the large number of microbial species constituting vertebrate gut microbiomes.

Host diet is considered one of the most relevant determinants of human gut microbiome variation[20,34–36]. The microbiome composition changes rapidly in response to new food choices, such as shifting from plant-based to animal-based diets[34,35,37] or changes in the protein to carbohydrate intake ratio[38]. Adding to this complexity, in humans the impact of diet on the microbiome is highly personalised[35]. While in vitro experiments have started to systematically disentangle the nutritional preferences of single human gut microbes[39], we are very far from achieving a coherent mechanistic picture of how bacterial dietary needs shape the microbiome and its capacity to influence the host. Given the large number of nutrients required by the host and the nutritional complexity of natural foods, identifying how single nutrients affect the microbiome and hence the host, remains a key challenge in current microbiome research.

Since the microbiome serves as a stable regulatory factor contributing to host physiology, it has been proposed that it should be resilient to dietary perturbations[40,41]. As in any other organism, gut microbes have very different nutritional requirements which at the individual species level, are largely defined by the biosynthetic capacities encoded at the genome level. The complex metabolic interactions within microbial communities can however profoundly alter the impact of nutrients on the physiology of the different species in the community[42]. In the context of the gut microbiome this could contribute to the emergence of dietarily resilient communities which would be stable even if the host diet lacks nutrients which are essential for specific members of the community. Identifying the mechanisms allowing gut microbe communities to overcome dietarily challenging conditions could therefore significantly expand our understanding of the conditions, in which the gut microbiome becomes susceptible to changes in host diet. This knowledge could also be used to guide the development of tailored interventions aiming at strengthening the resilience of gut microbe communities, thereby ensuring their continuous beneficial impact.

*Drosophila melanogaster* has emerged as a powerful, experimentally tractable system to identify the mechanisms by which gut microbes interact with the host to influence diverse traits ranging from metabolism, to growth and behaviour[17,20,43]. The adult *Drosophila* has a simple microbiome, typically containing <20 taxa, and mainly populated by species from the *Acetobacter* and *Lactobacilli* genera which can be cultivated in the laboratory and are amenable to genetic manipulations[44,45]. It is furthermore easy to generate and maintain germ-free animals as well as to reconstitute gnotobiotic animals with a predefined microbiome. Importantly, its powerful genetic and genomic toolset, the ability to perform large-scale, hypothesis-agnostic screens, and the availability of a chemically defined diet[46,47] make the fly an ideal system to identify core mechanistic principles governing diet–host–microbiome interactions[20,43].

In this study, we show that a syntrophic relationship between *Acetobacter pomorum* (*Ap*) and *Lactobacillus plantarum* (*Lp*), two abundant strains making up the fly microbiome, is at the base of their ability to suppress yeast appetite in flies deprived of eAAs. Using a chemically defined fly diet, we found that *Ap* can promote *Lp* growth in flies reared in media lacking isoleucine (Ile), an eAA for *Lp*. To explore the impact of host dietary conditions on the bacterial community, we adapt the fly holidic medium (HM) to be able to grow bacteria in vitro in high throughput. By combining this diet with isotope-resolved metabolomics, we show that the presence of *Lp* stimulates *Ap* to produce and excrete Ile and other AAs into the media. These AAs can then be used by *Lp* to grow in the absence of those in the diet. We furthermore identify lactate as the main contribution of *Lp* to the bacterial mutualistic relationship. As such, it is possible to substitute *Lp* with lactate and observe the same level of protein appetite suppression, showing that this metabolite is necessary and sufficient for modifying protein appetite in the presence of *Ap*. Interestingly, lactate is required for *Ap* to produce Ile, and using stable isotope labelled lactate, we show that it serves as a major precursor for the synthesis of AAs by *Ap*. These data provide clear evidence of a "circular economy" in which *Lp* derived lactate is used by *Ap* to generate AAs which allow *Lp* to proliferate and provide lactate to the community. Given that *Ap*, when supplemented with lactate, is sufficient to modify the behaviour of the host and can synthesise all eAAs, we tested the hypothesis that *Ap* modifies host behaviour by replenishing AAs in the malnourished host. We, however, provide multiple lines of evidence contradicting this hypothesis. We found that the bacteria only secrete extremely low levels of Ile compared to the levels required to alter behaviour in physiological conditions. We also show that the *Ap*/*Lp* community is beneficial for the host, allowing it to increase egg laying in malnourished females. This is most likely mediated

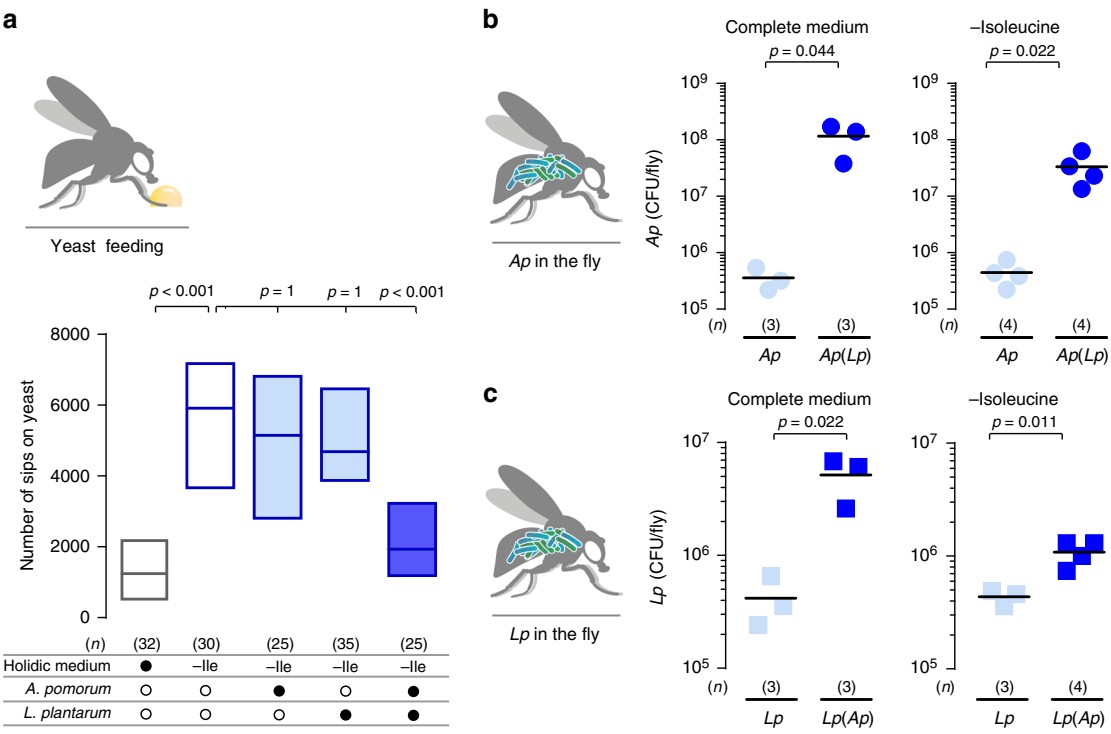

**Fig. 1 Ap/Lp biassociation reduces yeast appetite and promotes bacterial growth independently of dietary Ile content. a** Number of sips on yeast of flies kept in complete holidic medium or medium without Ile (-Ile) that were monoassociated (light blue) with Ap or Lp, or biassociated (dark blue) with both commensal species. Empty boxes represent germ-free conditions and filled boxes represent bacteria-associated conditions. Boxes represent median with upper and lower quartiles. n represents the number of flies assayed per condition. Significance was tested using a Kruskal–Wallis test with Dunn's multiple comparison test. Filled black circles represent a complete holidic medium or association with specific bacteria. Open black circles represent the absence of specific bacteria. **b, c** Number of Ap (circles, **b**) and Lp (squares, **c**) colony-forming units (CFUs) from extracts of monoassociated (light blue) and biassociated (dark blue) flies kept in complete holidic medium, or medium lacking Ile (-isoleucine). Ap and Lp label data points represent the number of CFUs detected in monoassociated flies and Ap(Lp) and Lp(Ap) the measurement of Ap or Lp in biassociated flies. Each data point represents the number of CFUs detected in independent fly extracts (n), and the black line represents the mean. Significance was tested using an unpaired two-tailed Student's t test. Source data are provided as a Source Data file.

by the commensal bacteria providing biomass (i.e. AAs); a mechanism which we show does not alter food choice. This allows us to functionally separate the beneficial effect of the community on egg laying from the effect on behaviour and strongly suggests that while bacterial biomass can improve egg laying, some other Ap-derived factor than bacterial produced eAAs influences food choice. This work demonstrates the importance of bacterial communities as the relevant explanatory unit necessary to understand how the gut microbiome impacts the host. We furthermore uncover the molecular basis of a circular metabolic cross-feeding relationship that supports the stability of a gut microbial community, making it resilient to the nutritional environment that the host may encounter. The resilience to dietary challenges allows the community to exert its beneficial impact on reproduction when the animal is malnourished. And importantly, it also ensures the continuous availability of metabolic precursors used by the behavioural effector species to alter brain function.

## Results

**Ap and Lp buffer yeast appetite in eAAs-deprived flies.** The impact of the microbiome on host physiology mostly emerges from the complex interaction of diet, gut bacteria, and host physiology[20]. The use of chemically defined (holidic) diets in combination with gnotobiotic animals are ideal tools to disentangle these complex interactions[8,47]. We pioneered the use of this approach to show that in mated *Drosophila melanogaster* females, the removal of any of the 10 eAA, induces a strong and

specific appetite for yeast (the main protein source for flies; Fig. 1a and Supplementary Fig. 1a) and that the fly microbiome alters the food preferences of the fly[8]. Here and throughout this study, we decided to focus on two eAAs, Ile and histidine (His), as proxies for all 10 eAAs. These two AAs represent very different chemical and biological classes of eAAs, with one being a branched-chain amino acid (BCAA), allowing us to cover a broad spectrum of biological activities. We used this framework to explore the interaction between dietary eAAs and the gut microbiome, and its effect on protein appetite. Remarkably, a community of two bacteria, consisting of Lp together with Ap, can significantly suppress the yeast appetite of flies deprived of eAAs (Fig. 1a and Supplementary Fig. 1a). Whereas, the presence of any of these bacteria alone cannot suppress protein appetite[8]. These results clearly show that the impact of the microbiome on feeding decisions relies on the presence of a minimal bacterial community consisting of only two members. This is in line with the common observation that many effects of the microbiome on host physiology and behaviour can only be understood at the level of microbial communities but not single bacteria[8,28–30].

**Syntrophy protects bacteria from dietary perturbations.** To mechanistically understand why the behavioural impact of the gut microbiome requires a community, we decided to start by testing the impact of diet and community composition on the titres of the different bacteria in the gut of the host. Diet is a potent modulator of the microbiome, and it is therefore likely that the composition of different fly diets may impact bacterial composition, and hence

　　　　　　　　　NATURE COMMUNICATIONS | https://doi.org/10.1038/s41467-020-18049-9

their effect on the host[48–51]. Especially as both *Ap* and *Lp* are auxotrophic for different nutrients including eAAs[52–54]. We therefore compared the internal bacterial load of flies inoculated with single cultures or co-cultures of *Ap* and *Lp* maintained for 3 days on a complete HM, or in media lacking Ile or His. In the tested flies, the number of both *Ap* and *Lp* cells was consistently higher in the co-culture conditions when compared to the bacterial load in flies inoculated with each bacterium alone (Fig. 1b, c and Supplementary Fig. 1b, c). This is in agreement with other reports showing synergistic effects in bacterial growth of *Acetobacter* and *Lactobacillus* species in *Drosophila*[55,56].

To test if the increase in *Ap* titres observed in the co-culture condition is sufficient to reduce protein appetite, we inoculated flies with a high dose of *Ap* ($10^9$ CFUs). When deprived of Ile flies with higher *Ap* titres did not show changes in protein appetite when compared to germ-free controls (Supplementary Fig. 2a). We also tested, if the bacteria need to be metabolically active or if the bacterial biomass was sufficient to change the protein appetite of the host. To ensure that we use a bacterial mass which approximates the one present in the gnotobiotic experiments we first measured the bacterial titres in the fly food vials containing HM lacking Ile (see Methods section for details). After 3 days of growth under this condition, *Ap* had reached a titre of $10.1 \times 10^9$ CFUs and *Lp* reached $13.1 \times 10^9$ CFUs. We added these amounts of heat-killed *Ap* and *Lp* to germ-free flies maintained on HM lacking Ile repeatedly over 3 consecutive days. In agreement with the outcome of the experiment where flies were treated with high doses of *Ap*, flies treated with high levels of dead bacterial biomass did not show a reduction in protein appetite (Supplementary Fig. 2b). This result strengthens previous findings that the bacteria must be metabolically active to alter behaviour[8] and highlights that a simple increase in biomass cannot explain how commensal bacteria alter food choice.

Interestingly, the increase in bacterial load when the microbes were co-cultured was observed both in the complete medium and the media lacking Ile or His (Fig. 1b, c and Supplementary Fig. 1b, c). This is expected for the condition in which we removed His from the diet, as both *Ap* and *Lp* should be able to synthesise this eAA[52,57,58]. However, given that *Lp* is theoretically unable to synthesise Ile it is unexpected that in flies maintained on media lacking Ile, the levels of *Lp* can increase (Fig. 1c). This clearly indicates that in the co-culture condition, *Lp* can grow in media lacking Ile, which given the predictions from the *Lp* genome and previous in vitro growth experiments should not be possible[52,54,57,58]. These data indicate that the presence of *Ap* allows *Lp* to overcome its Ile auxotrophy to remain metabolically active. The community is therefore able to buffer single bacteria from adverse effects of host diets lacking specific essential nutrients.

***Ap* allows *Lp* to overcome its isoleucine auxotrophy**. To carefully dissect the impact of the diet on bacterial growth and exclude the host as a confounding factor, we adapted the holidic fly medium to grow bacteria in in vitro liquid cultures (see Methods section). This also allowed us to perform bacterial dietary manipulations on a large scale while maintaining the bacteria in the same diet as we use to maintain the host. We first cultivated *Ap* and *Lp* alone in the liquid versions of the complete holidic fly medium or media lacking either Ile or His and assessed the growth of these bacteria over 3 days. After an initial rapid decrease in titres likely due to the transition from bacterial growth medium to HM, *Ap* grew to the same extent in complete medium, and in media lacking Ile or His (Fig. 2a and Supplementary Fig. 3a). This confirms the data obtained in the host and the known biosynthetic activities of this bacterium (Fig. 1b and

Supplementary Fig. 1b). It also shows that the liquid version of our HM is suitable for cultivating *Drosophila* commensal bacteria. Interestingly, in the in vitro situation, we did not observe a clear increase in growth in the co-culture condition when compared to the condition in which we grew *Ap* alone (Fig. 2 and Supplementary Fig. 3 vs. Fig. 1b, c and Supplementary Fig. 3). This strongly suggests that the overall increase in commensal bacteria proliferation observed in the host when they grow as a community is specific to the intestinal niche. This points to beneficial effects of the community, which are specific to the situation in which bacteria are growing in the animal gut, and highlights the strength of the in vitro culture system in isolating pure dietary effects from effects resulting from bacteria–host interactions.

While there was no effect of removing His from the medium on the growth of *Lp* alone (Supplementary Fig. 3b), *Lp* could not grow in the absence of Ile (Fig. 2b). However, when co-cultured with *Ap*, *Lp* was able to efficiently grow in media lacking this AA despite being auxotrophic for it (Fig. 2b). Intriguingly, both in the in vivo and in vitro situations (Figs. 1c and 2b) *Lp* seemed to grow to a lower titre in the –Ile situation when compared to the full medium, suggesting that while *Ap* can rescue the auxotrophy of *Lp*, it cannot do so efficiently. The ability of *Ap* to support *Lp* growth in liquid medium lacking Ile reproduces the observations made in the host situation. Our findings strongly suggest that *Ap* does so by providing Ile to *Lp*.

**Amino acids are synthesised by the bacterial community**. Our in vivo and in vitro bacterial growth data suggest that *Ap* would have to synthesise and secrete Ile to allow *Lp* to overcome the dietary lack of this eAA. To test this hypothesis, we decided to use stable isotope labelling to measure de novo synthesis and secretion of AAs from dietary glucose by the gut bacteria. First, we cultivated *Ap* alone in -Ile media containing uniformly labelled $^{13}C_6$-D-glucose to track the synthesis of $^{13}$C-Ile and other $^{13}$C-labelled AAs. Given our interest in the secreted fraction of AAs, we measured the amount of labelled AAs in the supernatant of the cultures using liquid chromatography–mass spectrometry (LC–MS). Clear differences in the pattern of isotopologue distributions between the *Ap* alone and the co-culture condition were already clearly visible at the level of the raw MS peaks of isoleucine (Supplementary Fig. 4). While *Ap* should be synthesising Ile to be able to grow in a medium lacking this eAA, we could not detect secreted, labelled Ile when we cultivated *Ap* alone for 24 h and only a modest increase after 48 h (only the $m + 2$ form; Fig. 3 left section). However, when *Ap* grew in a community with *Lp* for 24 h or 48 h, we could detect a substantial increase in the presence of the $^{13}$C-labelled $m + 2$ isotopologue as well as a minor increase in multiple other isotopologues ($m + 3$ to $m + 6$) of Ile (mean percent enrichment (MPE) at 24 h of $^{13}C = 36.65\%$ ± 3.32 (mean ± standard error of the mean (s.e.m.))) when compared to *Ap* single culture conditions (MPE = 0%) as well as the complete medium (Fig. 3 central section; MPE = 0%), and the –His situation (Fig. 3 right section; MPE = 0%). The low levels in secreted, synthesised Ile observed when the bacteria were grown in media that contained Ile (either complete medium or medium lacking His), suggests that the absence of dietary AAs stimulates the biosynthesis of this metabolite. Furthermore, when analysing other AAs we could see an overall increase in labelled and secreted AAs in the co-culture condition when compared to when *Ap* was grown in isolation (Supplementary Fig. 5). Interestingly, in the 24 h *Ap* culture, we also observed a clear overall decrease in the AAs provided by the HM when compared to the co-culture condition (unlabelled $m + 0$ form in Supplementary Fig. 3), which was mostly recovered at the 48 h time point. We interpret these data as meaning that when possible the bacteria consume

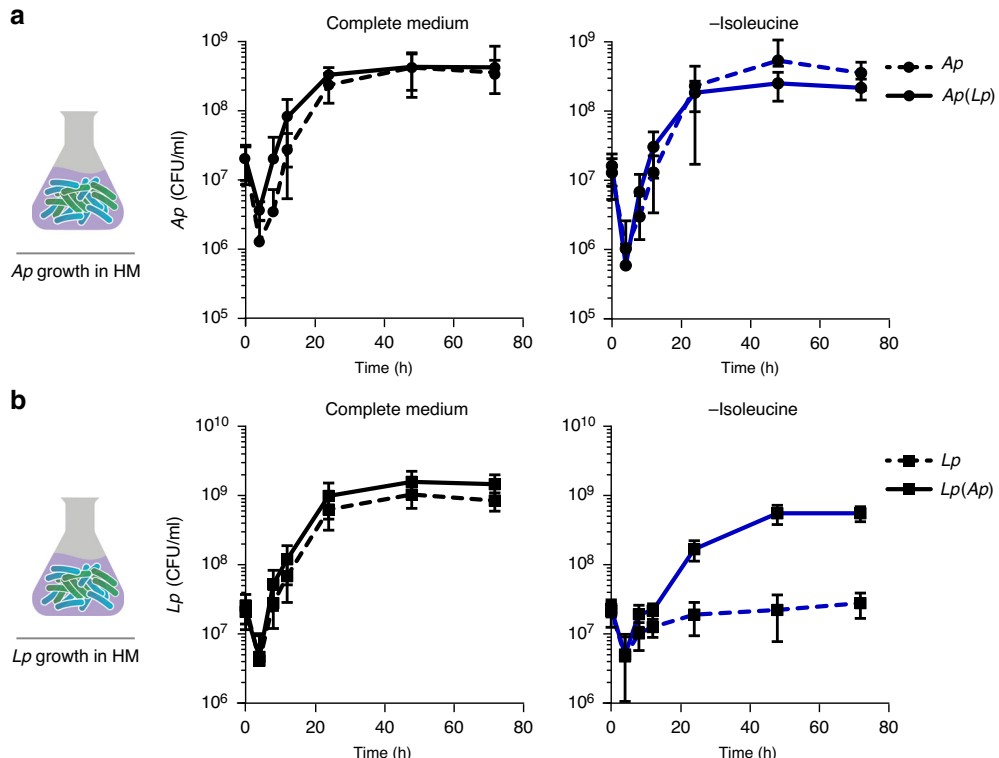

**Fig. 2 In vitro growth measurements show that *Lp* can overcome its auxotrophy for Ile when co-cultured with *Ap*.** In vitro growth curves of *Ap* (**a**) and *Lp* (**b**) in complete liquid holidic medium (black line) or in liquid medium without Ile (blue line) plotted as CFUs at different time intervals. Data points connected by dashed lines represent the number of CFUs detected in cultures of individual species (*Ap* or *Lp*) and the data points connected by continuous lines, the number of *Ap* (labelled as *Ap(Lp)*) or *Lp* (labelled as *Lp(Ap)*) CFUs in co-culture conditions. The number of CFUs per ml of culture was determined by cultivating liquid culture samples in selective media at the indicated time points. For each condition, seven independent bacterial cultures were sampled at 0 h, 24 h, 48 h and 72 h. Additionally, four cultures were sampled at 4 h, 8 h and 16 h. Each data point represents the mean CFUs from independent bacterial cultures (n = 7 for 0, 24, 48 and 72 h; n = 4, 8 and 16 h) and the error bars the standard deviation (s.d.). Source data are provided as Source data file.

AAs provided in the medium and that the increased ability of the bacterial community to synthesise AAs de novo decreases the need of the bacteria for the consumption of AAs in the diet. This interpretation is in agreement with previous observations that the growth of commensal bacteria decreases the amount of AAs in the medium[59]. Taken together, these results strongly suggest that the production of AAs by *Ap* and/or their accumulation in the medium is stimulated by the presence of *Lp*. In conclusion, these data show that *Ap* produces and secretes Ile and other AAs when growing in co-culture with *Lp*, providing the biochemical basis by which the bacterial community overcomes nutritional challenges posed by imbalanced host diets.

***Lp* contributes lactate to the bacterial syntrophy**. We have shown that *Ap* allows *Lp* to overcome its Ile auxotrophy by providing this limiting AA. Furthermore, our data clearly show that both the behavioural effect on the host, as well as the increase in available AAs, is a community effect depending on the presence of *Lp*. Therefore, *Lp* must be making a critical contribution to the community, both in the host as well as when growing in vitro. What could this contribution be? We have shown that both *Lp* and *Lactobacillus brevis* are interchangeable in their capacity to suppress yeast appetite when inoculated in co-culture with *Ap*[8]. Therefore, given that both are lactic acid-producing bacteria, one attractive hypothesis is that *Lp* provides lactate to *Ap*, which then is used as a metabolic precursor by this bacterium. Supporting this hypothesis, in the labelled metabolomics data, we

could detect significant amounts of lactate produced from glucose in the *Ap/Lp* co-culture condition (Supplementary Fig. 7).

To directly test if lactate production is necessary for the observed effect of the bacterial community on the behaviour of the host, we chose to genetically ablate lactate production in *Lp* and assess if this affected the ability of the bacterial community to suppress yeast appetite. We used an *Lp*^WCFS1 strain harbouring a deletion of the *ldhD* and *ldhL* genes (*Lp*^WCFS1Δ*ldh*), which had been shown to be important for lactate production in this *Lp* strain[60]. While the co-culture of *Ap* with the *wt Lp*^WCFS1 control strain strongly reduced the Ile deprivation-induced increase in yeast feeding, the *Lp*^WCFS1Δ*ldh* mutant strain failed to suppress yeast appetite (Fig. 4a). The failure of the *Lp*^WCFS1Δ*ldh* mutant strain to suppress yeast appetite could be compensated by adding back lactate to the medium, confirming the conclusion that lactate production by *Lp* is necessary for the community to alter the food choice of the host. These data clearly show that lactate production by *Lp* is necessary for the commensal bacteria community to alter food choice.

These results prompt the intriguing possibility that lactate production is the only critical metabolite provided by *Lp* for the commensal bacteria community to be able to exert its effect on host behaviour. To test this hypothesis, we assessed if lactate is sufficient to replace *Lp* in its ability to affect host behaviour in the context of the commensal bacterial community. As expected, flies in which *Lp* was removed from the bacterial community showed the same Ile deprivation-induced yeast appetite as germ-free animals (Fig. 4b). Strikingly, replacing *Lp* with lactate in the *Ap*

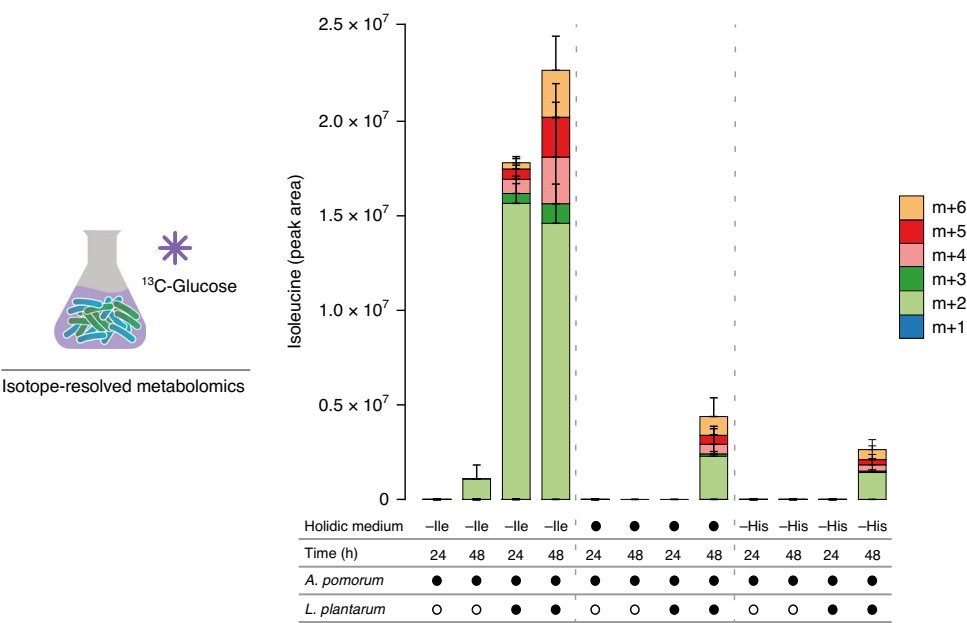

**Fig. 3 Ap synthesises and secretes isoleucine in the presence of Lp.** The stacked bars represent the relative amount of Ile isotopologues measured in the supernatant of liquid holidic medium of bacterial cultures grown in holidic medium lacking Ile (-Ile, left section), complete holidic medium (filled black circles, central section) or His (-His, right section) all containing uniformly labelled $^{13}$C-glucose. Heavy-labelled Ile isotopologues were measured using LC–MS in samples collected after 24 h or 48 h of bacterial growth and displayed as metabolite peak area. The number of heavy carbons incorporated per Ile molecule is indicated as $m + n$, where $m$ = molecular mass of the metabolite and $n$ = the number of $^{13}$C. Filled black circles represent the presence of specific bacteria in the culture. Open black circles represent the absence of specific bacteria. For each condition three independent bacterial cultures were sampled at 24 h and 48 h. The plots represent the mean amount of the detected isotopologues per condition and the error bars represent the s.e.m. Source data are provided as a Source Data file.

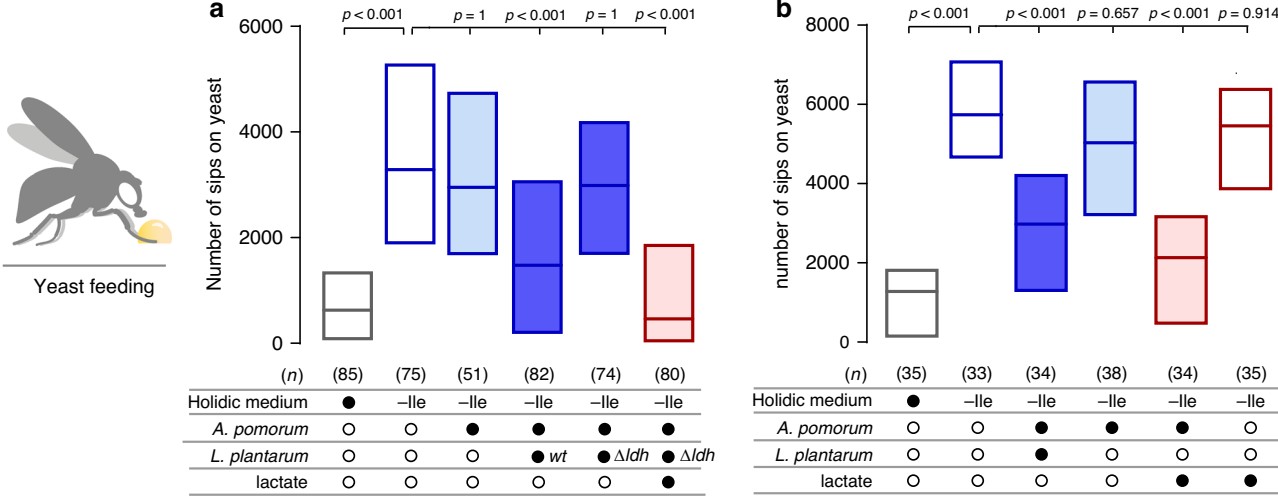

**Fig. 4 Lactate production by Lp is necessary and sufficient to reduce yeast appetite in the presence of Ap. a, b** Number of sips on yeast of either germ-free flies (empty boxes) kept in complete holidic medium (grey border boxes), or medium without Ile (-Ile, blue border boxes), or of flies that were monoassociated (light blue) with Ap, or biassociated (dark blue) with both Ap and Lp, or grown in medium containing lactate (burgundy). Filled black circles represent a complete holidic medium, or association with either Ap, wt or mutant Lp, or the presence of lactate in the medium. In **a** wt denotes the parental $Lp^{WCFS1}$ strain and $\Delta ldh$ the deletion mutant $Lp^{WCFS1\Delta ldh}$. Open black circles represent the absence of specific bacteria or lactate. Boxes represent median with upper and lower quartiles. n represents the number of flies assayed per condition. Significance was tested using a Kruskal–Wallis test with Dunn's multiple comparison test. Source data are provided as a Source Data file.

gnotobiotic animals lead to a potent suppression of yeast appetite, despite the host being Ile deprived for multiple days (Fig. 4b). This effect was not due to a direct, unspecific effect of lactate on the host as lactate alone (without Ap) did not affect yeast appetite.

These data together with the data from the lactate production mutant clearly show that lactate is the key metabolite provided by Lp allowing the bacterial community to alter food choice. Furthermore, these data strongly suggest that Ap is the main

microbial player altering host behaviour. And that *Lp* acts mainly as a provider of lactate, which could be required for *Ap* to synthesise the critical factors that drive the alterations in feeding behaviour.

**Lactate is used by *Ap* to produce amino acids.** What could be the metabolites synthesised from lactate by *Ap* required for the bacterial community to act on the host? Given that the synthesis

and secretion of AAs by *Ap* was increased by the presence of *Lp*, we wondered whether lactate could serve as a precursor for the production of these important nutrients. To test this hypothesis, we cultivated *Ap* in HM, in which we replaced *Lp* with uniformly labelled $^{13}C_3$-L-lactate. We tracked the synthesis of $^{13}C$ labelled AAs from this carbon source, normally provided by *Lp*, using LC–MS. Confirming our hypothesis, in the supernatant of *Ap* cultures grown in media lacking Ile, we found high levels of $m + 2$ isotopologues of $^{13}C$-Ile as well as modest levels of $m + 3$ to $m +$

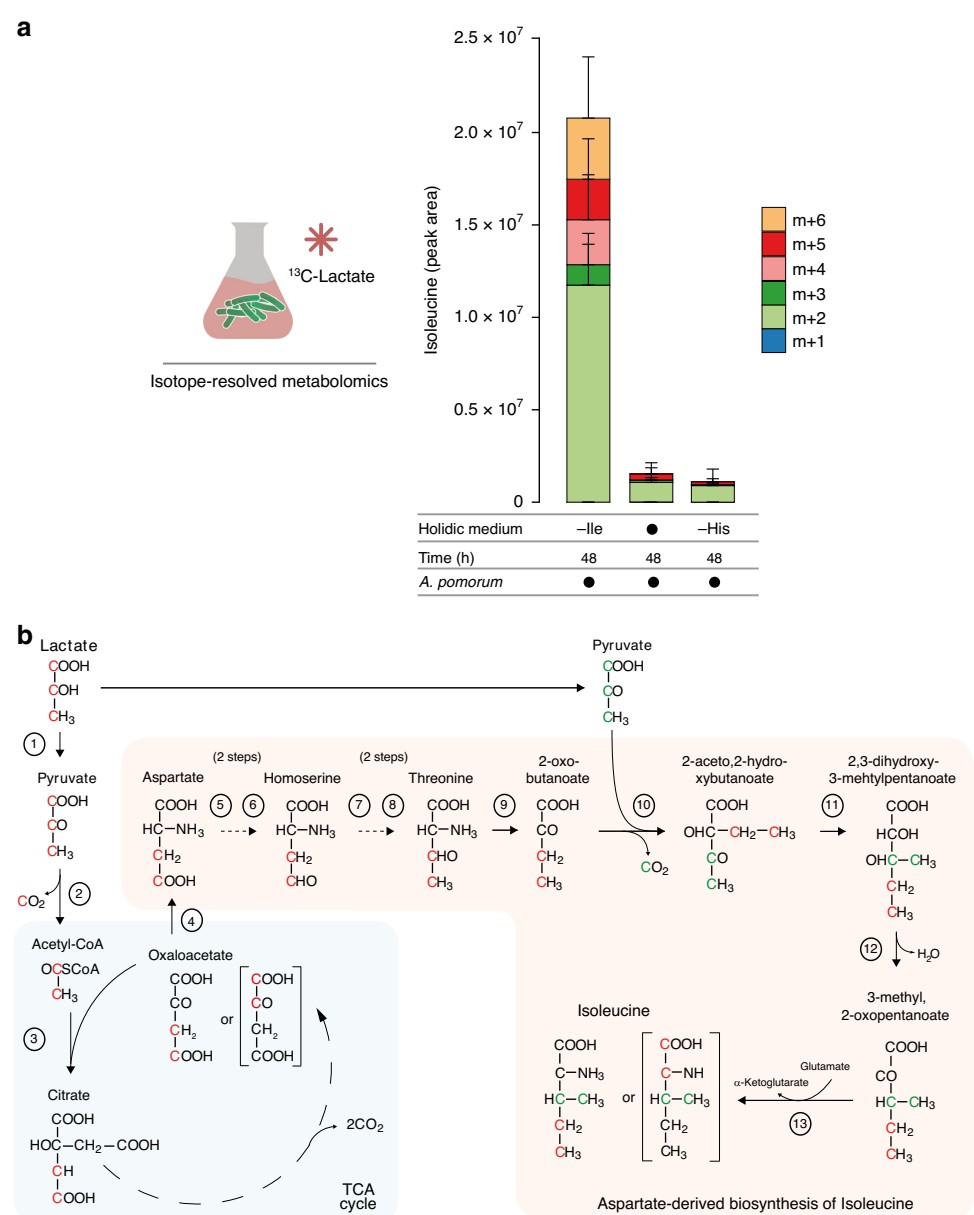

**Fig. 5 *Ap* synthesises Ile from lactate through an aspartate-derived biosynthetic pathway. a** The stacked bars represent the relative amount of Ile isotopologues measured in the supernatant of liquid holidic medium of bacterial cultures in complete holidic medium (filled black circle) or in medium without Ile (-Ile) or without His (-His) containing uniformly $^{13}C$ labelled lactate ($^{13}C$-lactate). Heavy-labelled Ile isotopologues were measured using LC–MS in samples collected after 48 h of bacterial growth and displayed as metabolite peak area. The number of heavy carbons incorporated per Ile molecule is indicated as $m + n$, where $m$ = molecular mass of the metabolite and $n$ = the number of $^{13}C$. Filled black circles represent the presence of specific bacteria in the culture. For each condition 3 independent bacterial cultures were sampled. The plots represent the mean amount of the detected isotopologues per condition and the error bars represent the s.e.m. Source data are provided as a Source Data file. **b** Predicted biosynthetic pathway for the synthesis of Ile from lactate in *A. pomorum*. Circled numbers correspond to the enzymes and reactions in Supplementary Table 1. The fate of lactate carbon atoms incorporated through the TCA cycle is indicated in red, while the pyruvate (derived from lactate) carbon atoms incorporated by enzyme 10 are labelled in green and red. Note that the resulting Ile molecule can contain 2 or 4 labelled carbon atoms depending on the labelling of the lactate molecules. Continuous arrows represent direct reactions; dashed arrows represent two or more consecutive reactions.

6 isotopologues (MPE = 35.95% ± 12.87 (mean ± s.e.m.); Fig. 5a). The accumulation is especially striking when compared to the levels found in the complete medium (MPE = 0.08% ± 0.08) or the medium lacking His (MPE = 0.05% ± 0.05). Remarkably, the labelling pattern of Ile was almost identical to the one detected in the *Ap/Lp* co-culture using $^{13}C_6$-D-glucose (Fig. 3). Furthermore, we could detect synthesis from lactate of all AAs that were previously detected as being produced from glucose in *Ap/Lp* cocultures. Similarly to Ile the labelling pattern of these AAs looked almost identical to the one observed using labelled glucose (Supplementary Fig. 8), highlighting the similarity between the co-culture condition and the lactate *Ap* condition. Our results show that *Ap* synthesises multiple AAs from lactate and agree with earlier reports that lactate can serve as an important precursor for AA synthesis in other *Acetobacteraceae*[61]. Altogether, these results show that *Ap* uses lactate to synthesise secreted AAs. In the host, *Lp*-derived lactate is therefore very likely to be used by *Ap* to synthesise AAs, which are then used by *Lp* to grow and produce lactate, allowing the community to overcome deleterious dietary conditions. The syntrophic relation between *Ap* and *Lp* therefore ensures the dietary stability of the community, while also ensuring a constant flux of lactate to *Ap*, enabling this bacterium to exert its function on host behaviour.

**Ap uses an aspartate derived pathway to synthesise Ile.** What is the metabolic pathway used by *Ap* to synthesise Ile? While lactate has been proposed to be an important precursor for AA synthesis in *Acetobacteraceae* the detailed metabolic pathway by which it does so remains elusive. We combined genome-based pathway reconstruction (Supplementary Table 1), with a detailed analysis of the isotopologue distributions observed in the glucose and lactate labelling experiments, to deduce the most likely biosynthetic pathway, by which this bacterium transforms lactate into Ile (Fig. 5b). The derived pathway proposes that lactate can either be fed into the TCA cycle after conversion into pyruvate to be transformed into oxaloacetate, aspartate and finally isoleucine. Or that it can be directly fed into the aspartate-derived biosynthetic pathway of Ile. This reconstruction is supported by multiple observations emerging from our metabolomics analysis. First, the most abundant form of labelled Ile is the $m + 2$ form, which suggests that lactate must undergo a decarboxylation step before it is used to synthesise Ile. This is indeed what our pathway suggests. Moreover, we also observed a significant synthesis of aspartate in -Ile condition at 48 h (Supplementary Fig. 5), while observing a significant consumption of unlabelled ($m + 0$) aspartate in the medium (Supplementary Figs. 6 and 9). This suggests that the biosynthesis of isoleucine is likely to be through the proposed pathway using aspartate as an intermediate metabolite. Obviously, this does not mean that Ile cannot be synthesised from other precursors than lactate. Importantly, our annotation of the enzymes from the genome of *Ap* suggests that all of these enzymes and reactions required for this pathway are present in *Ap* (Supplementary Table 1). Together with the observed changes in mass distributions and metabolite abundances, this strongly supports our conclusion that this pathway is the most likely mechanism for Ile synthesis from lactate by *Ap*. A final confirmation will have to nevertheless be provided by genetic studies using mutants in key enzymes of this pathway.

**Bacterial AAs are unlikely to reduce yeast appetite.** So far, our results show that *Ap* sustains *Lp* growth in vivo and that in vitro, in the absence of Ile and either in the presence of *Lp* or lactate, *Ap* increases the synthesis and secretion of AAs. It is widely accepted that gut microbes can supply eAAs to different host organisms, insects and humans included[62,63]. Our data would therefore be compatible with a simple model, in which *Ap* synthesised AAs would not only be required to ensure the growth of *Lp*, but would also act on the host to suppress yeast appetite. If this were the case, the amount of AAs provided by the bacterial community should be comparable with the amount of dietary AAs sufficient to suppress protein appetite in the axenic animals. We therefore first decided to compare the amount of Ile secreted by the bacterial community with the amounts measured in the complete fly medium, which we know efficiently suppresses yeast appetite. While in the Ile dietary deprivation situation we could detect de novo synthesised Ile in the co-culture as well as in the *Ap* culture supplemented with lactate (Figs. 3 and 5a), the amount of secreted Ile in the co-culture condition is 1/85th of the amount present in the complete HM (Fig. 6a).

To identify the amount of Ile required to suppress yeast appetite, we next titrated the concentrations of Ile in the HM and tested the feeding behaviour of axenic flies maintained on these diets. Strikingly, a diet with 25% of the total concentration of Ile did not lead to a significant suppression of yeast appetite, suggesting that this amount of Ile is not sufficient to significantly alter the behaviour of the fly (Fig. 6b). Only the addition of at least 50% of the full Ile concentration led to a complete suppression of protein appetite. This shows that commensal bacteria would have to provide relatively high amounts of dietary Ile (between 25 and 50% of the original amounts in the HM) to directly suppress yeast appetite. Our in vitro measurements, however, suggest that the bacterial community secretes ~100 times lower amounts of Ile than the ones required to suppress yeast appetite (Fig. 6a). This makes it unlikely that the amount of AAs secreted by the bacterial community is sufficient to suppress protein appetite.

We decided to back this conclusion using a different physiological readout for AA availability. In *Drosophila*, dietary eAAs are the main rate-limiting nutrients required for egg production[8,47,64]. As such, egg laying can be used as an almost linear readout for the physiological availability of eAAs[46]. We had shown that commensal bacteria can increase egg laying in eAA deprived animals[8]. Indeed, we observed that Ile deprivation led to a drastic decrease in egg laying in germ-free females, which was mildly rescued in females harbouring an *Ap/Lp* community (Fig. 6c). This can be interpreted as the community providing AAs to the host, which it then uses for producing eggs. It is however important to note that the rate of egg laying is still very low when compared to what would be expected on a diet containing between 25 and 50% of the original Ile diet[47]. We have shown earlier that the biomass (i.e. mostly proteins) provided by the bacteria cannot reduce protein appetite (Supplementary Fig. 2b). To test if the biomass could nevertheless explain the increase in egg laying, we measured egg production in germ-free mated females supplemented with heat-killed *Ap* and *Lp*. Indeed, the addition of the metabolically inactive bacterial biomass was sufficient to significantly increase egg laying in Ile deprived females (Fig. 6c). This strongly suggests that the Ile produced in the syntrophic community is sufficient to alter egg laying while not affecting behaviour, allowing us to functionally separate the impact of the microbiome on behaviour and reproduction. The ability of the dead bacteria to rescue egg laying in an –Ile situation, supports our finding that the syntrophic community produces Ile but that the amount provided by *Ap* is on its own not sufficient to alter feeding behaviour. This strengthens the evidence that bacterially synthesised AAs are unlikely to contribute to the suppression of yeast appetite.

Finally, we tested if we could detect an increase in free AAs in AA deprived flies with a bacterial community. For this, we performed metabolomics on isolated heads of germ-free and gnotobiotically bacterially reconstituted females (to avoid the

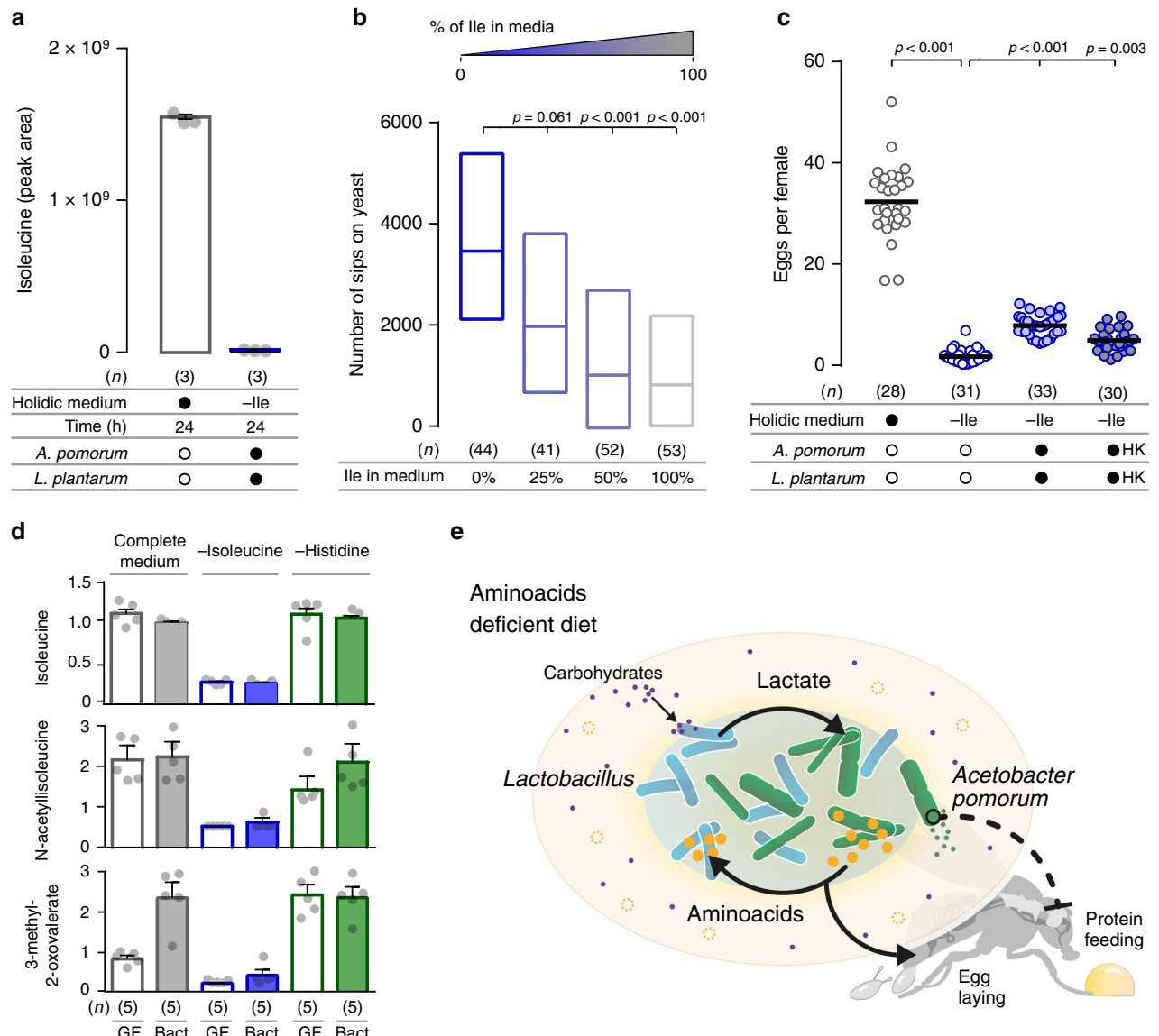

**Fig. 6 Bacterial amino acid supply is unlikely to explain the effect of gut bacteria on behaviour. a** Relative amount of total (labelled plus unlabelled) Ile determined by LC–MS in complete holidic medium without *Ap* and *Lp* and medium lacking Ile with *Ap* and *Lp*. Grey circles represent measurements in independent bacterial cultures (*n*), the column represents the mean, and error bars the s.e.m. **b** Number of sips on yeast of flies maintained in holidic medium lacking Ile (0%) or containing 25%, 50% or 100% of the Ile amount in complete medium. *n* represents the number of flies assayed per condition. Boxes represent median with upper and lower quartiles. **c** Each circle in the plot represents the number of eggs laid/female in single assays (*n*) and the line represents the mean. Females were fed complete holidic medium or medium lacking Ile treated with living or heat-killed (HK) bacteria. **d** Bars represent the relative abundance of Ile and Ile degradation metabolites in heads of germ-free flies (GF) and flies inoculated with commensal bacteria (Bact) and maintained in either complete medium, or medium lacking Ile or His. Grey circles represent measurements of metabolites in independent extracts of heads of flies (*n*), the column represents the mean and error bars the s.d. **e** Schematic describing the metabolic interactions between *Ap* and *Lp* and their potential effects on the bacterial community and the host. The absence of eAAs in the diet is represented as dashed circles. Black arrows represent the flow of *Lp*-produced lactate and its utilisation by *Ap* to produce AAs. These AAs are then used by *Lp* when dietary AAs are scarce. These are necessary for the beneficial effect of bacteria on egg laying. The black, dashed inhibitory arrow indicates the effect of *Ap* in suppressing host's protein appetite through a yet undetermined lactate-dependent mechanism. In **a**, **c** black filled circles represent a complete holidic medium or association with bacterium. Open black circles represent the absence of bacterium. **b**, **c** Significance was tested using a Kruskal–Wallis test with Dunn's multiple comparison test. Source data are provided as a Source Data file.

confounding contribution of reducing the number of eggs by eAA deprivation). Supporting previous results[8], in axenic conditions the removal of either Ile or His drastically reduced the levels of the corresponding free AAs (Fig. 6d and Supplementary Fig. 10). This finding is supported by the observed concomitant reduction in the levels of multiple metabolites derived from these AAs, such as *N*-acetylisoleucine, 3-methyl-2-oxovalerate, *N*-acetylhistidine and imidazole lactate (Fig. 6d and Supplementary Fig. 10). In

agreement with all our results, the presence of gut bacteria did not rescue this reduction in free amino acids and corresponding derived metabolites (Fig. 6d and Supplementary Fig. 10). The levels of the deprived eAAs and their metabolites in the fly remained very low in both the flies with and without gut bacteria. These measurements made in the host further support the conclusion that gut bacteria are unlikely to act on host behaviour by rescuing the levels of free AAs in the animal.

Overall our data are in agreement with a model, in which *Lp* and *Ap* grow as a community in the fly, where they engage in a syntrophic interaction buffering them from adverse host dietary conditions (Fig. 6e). *Lp* provides lactate to *Ap*, which it uses to synthesise and secrete AAs. This ensures *Lp* growth, even in detrimental dietary conditions in which limiting AAs are missing, as well as an increase in egg production in malnourished flies. It also ensures the constant flux of lactate, which provides the necessary fuel for *Ap* to synthesise the molecules that alter choice behaviour (Fig. 6e), which according to our data are unlikely to be proteogenic AAs. Syntrophic relations between gut bacteria could therefore be a common theme among gut bacterial communities, generating metabolic cycles that buffer them from suboptimal host dietary conditions, and allowing them to generate a constant flux of metabolites acting on the host.

## Discussion

Identifying the mechanisms by which metabolic exchanges shape diet-microbiome–host interactions is key to understanding how gut microorganisms alter the physiology and behaviour of the host. Both the food choice of animals and their microbiome are altered by changes in diet. How one mechanistically relates to the other is currently poorly understood. We had shown that two bacteria abundant in the fly microbiome (*Ap* and *Lp*) act together to suppress the protein appetite of AA deprived animals[8]. Here we show that they need to act as a community to establish a syntrophic relationship that enables them to overcome drastic nutritional limitations generated by imbalanced host diets. While most studies on the impact of host diet on the microbiome emphasise the ability of diet to alter the microbiome[20,34,35,49,50], our work highlights a different facet of diet-microbiome relationships: how metabolic interactions within the microbiome allow the gut microbiome to become resilient to changes in the host diet. Key is the ability of multiple microbes to establish communities, in which syntrophic relationships allow specific members of the community to overcome auxotrophies for specific nutrients. While widely studied in the context of microbial ecology[33,42,65], less is known about the metabolic interactions shaping gut microbes and their importance in how they act on the host. This is especially the case in humans, where we have just started analysing the nutritional preferences and metabolic idiosyncrasies of single members of the human gut microbiome[39]. While it has been proposed that microbial metabolic interactions might be key for the generation of specific effector metabolites such as GABA or serotonin[20,29,66], we show that an important aspect of gut microbial communities is their resilience towards dietary perturbations, thereby ensuring a stable and constant impact of the microbiome on the host.

Lactic acid bacteria are known to often co-occur with other microorganisms in a variety of natural niches[42,67]. One such niche is the adult *Drosophila* gut, where *Acetobacteraceae* and *Lactobacilli* are often found together[45,68–70]. The co-occurrence of bacteria from these two genera in the fly has been shown to increase the propagation of both bacteria and to contribute to the impact of the microbiome on physiological traits of the host[28,55,56]. Although lactate has been proposed to play a role in these interactions[56], the mechanisms that promote and sustain bacterial growth in co-cultures and how they modify the host remain largely unexplored. *Lactobacilli* lack many key genes required for the synthesis of different essential nutrients, such as AAs and vitamins[54,57,58,71]. Conversely, their genome encodes an unusually large repertoire of transporters, highlighting their ability, if not requirement, to take up nutrients from their environment[57,72]. Using a chemically defined diet[46], we show that *Lp*, a bacteria auxotrophic for Ile, can proliferate in vitro via the

uptake of Ile produced by *Ap*. Moreover, the cross-feeding of this eAA, most likely, occurs also in vivo, since *Lp* levels are higher in biassociated than in monoassociated flies deprived of Ile, and *Lp* is necessary with *Ap* for altering feeding behaviour and egg laying in these flies. Microbial cross-feeding has been shown to allow intra- and inter-species exchange of several nutrients, including AAs, through its secretion to the media or via bacterial nanotubes[65,73,74]. The *Ap*–*Lp* interaction is however not uni-directional as has been shown for example for yeast-*Lactobacilli* interactions[75]. The production and secretion of AAs by *Ap* depends, or is strongly enhanced, by the presence of *Lp*. Concomitantly, this interaction also spurs the growth of *Ap* as this bacterium grows better in the fly in the presence of *Lp*. This positive effect of *Lp* on AA synthesis by *Ap* is best explained by the preferential use of lactate by *Ap* for the production of AAs[61], which coincidentally is one of the main metabolic by-products produced and secreted by *Lp*. These metabolic interactions within the *Ap*/*Lp* community allow these two bacteria to create a "circular economy", in which they both optimally use the available nutritional resources provided by the host diet, allowing them both to overcome detrimental host diets and boosting their metabolic output.

Our results also strongly suggest that one or multiple molecules derived from *Ap* lactate metabolism are likely to be the effectors altering feeding behaviour in flies co-inoculated with *Ap* and *Lp*. This conclusion is based on the observation that lactate can fully substitute *Lp* to suppress yeast appetite in *Ap*-monoassociated flies deprived of Ile. One straightforward hypothesis is that *Ap*, an autotrophic strain, synthesises and provides eAAs to the host, suppressing protein appetite. In fact, there is evidence that gut microbes can supply significant amounts of essential nutrients, like AAs, to the host including humans[62,63,76,77]. However, we find that the bacteria produce and secrete low amounts of AAs compared to those existing in the complete HM, which has been optimised to promote egg laying and lifespan of the fly (two orders of magnitude less). This significantly weakens the hypothesis that *Ap* alters yeast appetite by supplying enough eAA to the host, so that it can compensate for the dietarily absent eAA. Especially considering that our psychometric measurements indicate that concentrations of 50% or more of the Ile concentrations in the media are necessary to significantly suppress protein appetite. This strongly suggests that the gut bacteria would need to produce at least that amount of AAs to alter protein appetite via these metabolites, which is not compatible with our in vivo and in vitro measurements. Finally, the mono-association of malnourished flies with high titres of *Ap* or high titres of dead, metabolically inactive *Ap*/*Lp* mixtures does not alter protein appetite. Interestingly, the bacterial biomass on its own can increase egg laying, a physiological process that is profoundly dependent on AA availability. This supports our data that the syntrophic interaction allows the bacteria to generate AAs and that even though they are not sufficient to change feeding behaviour, the produced AAs significantly but modestly rescue egg laying in malnourished females. Despite this modest contribution of AAs by the bacteria, we could not detect a rescue of the free AAs, which had been removed from the diet, in the heads of flies associated with bacteria suggesting that the modest amount of AAs available for the fly get funnelled into supporting egg laying. All these results together with earlier published data[8], fail to support the straightforward hypothesis that the bacteria act on yeast appetite by producing enough eAAs to replace the effect of the dietary removal of specific eAAs.

It is however important to note that in the physiological *Ap*/*Lp* biassociation situation, the presence of the microbiome is beneficial for the animal as it allows the fly to lay more eggs in an eAA deprived diet situation. This might sound counterintuitive to the

above-mentioned arguments as egg laying requires building blocks. But this apparent contradiction can be easily resolved by our finding that egg laying and protein appetite can be functionally separated. The modest improvement in egg laying is likely to result from the use of the increased bacterial biomass in the biassociation situation, which is sufficient to support a modest amount of egg laying while not being sufficient to suppress yeast appetite. In a malnutrition setting the $Ap/Lp$ community is hence beneficial for the fly as it allows it to maintain egg laying. Importantly, this benefit requires the ability of the community to grow in a diet lacking AAs for which $Lp$ is auxotrophic. The ability of the bacterial community to withstand dietary perturbations and produce AAs is therefore key for its beneficial effect on the adult host.

If the bacterial community does not act on food choice behaviour by maintaining a high level of eAAs in the host, how does it then modify behaviour? The here presented data support the hypothesis that it is one or multiple, lactate-derived, $Ap$-generated metabolite (s) which modulate the feeding behaviour of the fly. This fits with our data suggesting that the gut bacteria need to be metabolically active to modify food choice[8]. Gut bacteria can contribute with a plethora of small metabolites on concentrations comparable to those administered in drug doses ($10 \mu M$–1 mM)[20,78]. These include neuroactive substances such as GABA and serotonin[79,80]. Furthermore, recent studies identified multiple bacterial metabolites acting as G-protein-coupled receptors (GPCRs) agonists, which have the potential to affect host physiology, including the nervous system[81,82]. The reduction of the complexity of the behaviourally active community to one bacterium ($Ap$) and the identification of lactate as a likely precursor for the generation of the neuroactive metabolites, allows a targeted focus to identify the precise mechanisms by which the gut microbiome alters food choice. A combination of metabolomics approaches, including stable isotope-resolved metabolomics, and bacterial genetics, including unbiased genetic screens, should allow for the identification of the exact molecular mechanisms underlying the changes in behaviour.

Our work identifies the mechanisms that sustain a syntrophic relation between two abundant species of bacteria found in the fly's microbiome, proposes the metabolic pathway used by an Acetobacteraceae to synthesise a key AA using lactate as a precursor, and identifies the species that drives the alteration in behaviour observed in the host. We also show that this mutualistic relationship buffers the effect of dietary restrictions in both, the microbiome and the host. Diet is an essential, dynamic, and highly diverse environmental variable deeply affecting several aspects of behaviour, including food choice[8,10–12,83], as well as the microbiome, which has been shown to play a critical role in human behaviour, including through the metabolism of nutrients and drugs[20,30,77,84–86]. Our study highlights the importance of metabolic interactions among different species of gut bacteria in shaping the outcome of diet on the microbiome and its impact on host physiology and behaviour. Given the numerical complexity of the human microbiome and its high inter-individual heterogeneity[87] disentangling such relationships mechanistically is a daunting task. The ability to combine extremely precise dietary manipulations using a holidic diet, as well as microbial, genetic and molecular perturbations with detailed behavioural, physiological and metabolic phenotyping in high throughput makes Drosophila an ideal system to identify mechanisms by which gut bacterial communities act on the host. Importantly, given that the key molecular and physiological regulatory mechanisms are conserved across phyla, the microbiome mechanisms identified in invertebrates have been shown to be translatable to vertebrates[88]. Our study shows the potential of using Drosophila to mechanistically disentangle the influence of diet on microbiological communities and identify the individual contributions of bacterial species on host behaviour and brain function. The identified molecular and metabolic strategies can then be harnessed to explore similar mechanisms in vertebrates, including humans, providing an attractive path for the efficient mechanistic dissection of how gut microbes act on the host across phyla.

## Methods

**D. melanogaster husbandry and dietary treatments.** All experiments were performed with either axenic or gnotobiotic mated $w^{1118}$ female flies. Flies were reared under controlled conditions at 25 °C, 70% humidity, and 12 h light/dark cycle. Axenic fly stocks were generated as previously described in Leitão-Gonçalves et al.[8] (see also dx.doi.org/10.17504/protocols.io.hebb3an) and maintained on high yeast-based medium (highYBM) (per 1.1 litre of water: 8 g agar [NZYTech, PT], 80 g barley malt syrup [Próvida, PT], 22 g sugar beet syrup [Grafschafter, DE], 80 g cornflour [Próvida, PT], 10 g soy flour [A. Centazi, PT], 60 g instant yeast [Safinstant, Lesaffre], 8 ml propionic acid [Argos] and 12 ml nipagin [Tegosept, Dutscher, UK] [15% in 96% ethanol]) laced with antibiotics (50 µg ml⁻¹ tetracycline, 50 µg ml⁻¹ kanamycin, 50 µg ml⁻¹ ampicillin, and 10 µg ml⁻¹ erythromycin). The absence of bacteria was assessed regularly by grinding flies in sterile 1x PBS and spreading the suspension on LB, MRS or Mannitol plates. For the experiments described in this study axenic fly cultures were set in sterile highYBM without antibiotics using 6 females and 4 males per vial, to guarantee a homogeneous density of offspring across the different experiments, and left to develop until adulthood in this media. Holidic medium (HM) without preservatives and with an optimised AA composition (FLYAA) was prepared as previously described in Leitão-Gonçalves et al.[8] (see also dx.doi.org/10.17504/protocols.io.heub3ew). HM lacking isoleucine or histidine was prepared by simply omitting the corresponding AAs. A 300-g l⁻¹ lactic acid solution (pH 4.0-4.5) was prepared from a 90% (w/w in H₂O) ᴅ-lactic acid solution (Sigma-Aldrich, #69785) using 10 M NaOH, and used to prepare HM supplemented with lactate at 20 g l⁻¹. Flies were exposed to the different dietary treatments using the protocol described in Leitão-Gonçalves et al.[8] and dx.doi.org/10.17504/protocols.io.hhcb32w where axenic or gnotobiotic 1–5-day-old flies were used. Briefly, 16 females and 5 males were placed in highYBM-containing vials (without preservatives), transferred to fresh highYBM after 48 h and transferred, after 24 h, to the different HM (sterile or bacteria-treated media) where they were maintained for 72 h before any indicated assay.

**Bacterial species and generation of pre-inocula.** The following bacteria strains were used in this study: Lactobacillus plantarum^NC8 [89], Lactobacillus plantarum^WCFS1 [60], Lactobacillus plantarum^WCFS1Δldh [60] and Acetobacter pomorum[70]. To generate the five bacteria community used for the fly head metabolomics measurements the following bacterial strains were also used: Lactobacillus plantarum^WJL [70], Lactobacillus brevis^EW [70], Commensalibacter intestini^A911T [70], and Enterococcus faecalis[90]. All Lactobacillus species were cultivated in 10 ml MRS broth (Fluka, #38944) as static cultures in 14 ml culture tubes (Thermo Fisher Scientific, #150268) at 37 °C for 24 h. A. pomorum and C. intestini^A911T were cultivated in mannitol (3 g l⁻¹ Bacto peptone [Difco, #0118–17], 5 g l⁻¹ yeast extract [Difco, #212750], 25 g l⁻¹ ᴅ-mannitol [Sigma-Aldrich, #M1902]) at 30 °C for 48 h with orbital agitation (180 rev. min⁻¹). C. intestini^A911T was cultured in 20 ml of medium in 50-ml tubes (Falcon), while A. pomorum was cultured in 200 ml of medium in 500-ml flasks. E. faecalis was cultured in 200 ml of liquid LB medium (Sigma-Aldrich, #L3022) in 500-ml flasks at 37 °C for 24 h at 220 rev min⁻¹.

**Generation of axenic and gnotobiotic flies.** Axenic flies were generated as previously described in Leitão-Gonçalves et al.[8] and dx.doi.org/10.17504/protocols.io.hebb3an. To prepare mono- and biassociated flies with A. pomorum and Lactobacilli strains, vials containing HM media were inoculated with the desired bacterial species before flies were transferred. The number of CFU added per vial was as follows: L. plantarum^NC8, 7.2 × 10⁷; L. plantarum^WCFS1, 5.0 × 10⁷; L. plantarum^WCFS1Δldh, 5.6 × 10⁷; and A. pomorum, 9.5 × 10⁷. In the particular case of the gnotobiotic flies used to measure the metabolites in the fly's head the bacterial community consisted of: L. plantarum^WJL (6.4 × 10⁷ CFU), L. brevis^EW (5.31 × 10⁶ CFU), C. intestini^A911T (9.04 × 10⁷ CFU), A. pomorum (9.5 × 10⁷ CFU) and E. faecalis (1.11 × 10⁸ CFU). The volume of culture containing the appropriate number of CFUs was centrifuged (5000×g, 10 min), the pellet washed three times with 1xPBS, and the cells resuspended in 50 µl of 1xPBS (per vial). The same volume of sterile bacterial media was centrifuged and the PBS resulting from washing the tube three times was used as a control. Fifty microliters of cell suspension were added to each fly culture vial, and those allowed to dry for 2 h before transferring flies.

**Assessment of bacterial CFUs.** The selection and assessment of the number of A. pomorum or L. plantarum^NC8 CFUs in flies co-inoculated with two bacterial species or in liquid co-cultures was done by plating samples in MRS media either supplemented with kanamycin (50 µg l⁻¹) or ampicillin (10 µg l⁻¹) for the selection of L. plantarum^NC8 or A. pomorum, respectively. MRS plates supplemented with kanamycin were incubated at 37 °C, while plates supplemented with ampicillin were incubated at 30 °C. Samples of monoassociated flies were plated in MRS and

incubated at 30 °C or 37 °C for *A. pomorum* or *L. plantarum*[NC8], respectively. All samples were plated using an easySpiral® (InterScience, #412000) plater and the number of CFU counted with an automatic colony counter Scan® 500 (Interscience, #436000).

**Assessment of the bacterial load of flies**. Flies were washed in 70% ethanol, to remove bacteria adhering to the fly cuticle, and further washed twice with sterile 1x PBS. The flies were homogenised using 50 μl of 1x PBS per fly and the homogenates serially diluted and plated in an appropriate media for the selection of the different bacteria. The number of CFU per fly was measured in at least three independent biological samples of eight flies.

**Treatment with heat-killed bacteria**. The bacterial load (CFUs) in vials containing HM lacking Ile, containing flies that were biassociated with *A. pomorum* and *L. plantarum*[NC8], was determined on the day of the behavioural assay (3 days after bacterial inoculation). An average of $1.01 \times 10^9$ ($\pm4.21 \times 10^8$) *A. pomorum* CFUs and $1.31 \times 10^9$ ($\pm8.46 \times 10^8$) *L. plantarum* CFUs were detected. The bacterial load was determined in three independent experiments.

To generate vials containing dead bacterial biomass, pre-inocula of the two bacteria were prepared as described in the "Bacterial species and generation of pre-inocula" section. A number of cells corresponding to three times the previously measured average number of bacteria measured in HM vials was harvested by centrifugation. The cells were washed twice with 20 ml of 1x PBS and resuspended in 150 μl of 1x PBS. This cell suspension was autoclaved for 30 min at 121 °C as previously described[91]. During the 3 days of deprivation in HM lacking Ile, flies were provided every day with 50 μl of the autoclaved-cell suspension (heat-killed bacteria; HK). To add the HK cell suspension at days 1 and 2, experimental as well as control flies were transferred to empty vials, while in the experimental condition the HK cell suspension was added and spread on the corresponding vial surface. The vials were left to dry for 20 min and then all flies were transferred back to the HM-containing vials. This ensured that all flies underwent the same handling procedure.

**flyPAD assays**. Food choice experiments were performed using the flyPAD as described in Itskov et al.[92]. The food preferences of single gnotobiotic flies, maintained in complete HM or deprived of single AAs, were tested in an arena with food patches containing 1% agarose mixed with either 10% yeast or 20 mM sucrose. The flies were allowed to feed for 1 h and the number of sips per animal was calculated using the previously described flyPAD analysis algorithms in Itskov et al.[92]. Briefly, the flyPAD data were acquired using the Bonsai framework[93] (versions 2.1.4; 2.3; 2.3.1), and analysed in MATLAB (version 8.2.0.701) using custom-written software, as described in Itskov et al.[92]. Non-eating flies (those flies having less than two activity bouts per assay) were excluded from the analysis. To test if glucose (100 mM) can replace sucrose (50 mM) as a carbon source in the HM, the feeding behaviour of axenic and biassociated flies was tested using the flyPAD. Axenic flies fed the glucose-based HM diet exhibited the same increase in yeast appetite when deprived of isoleucine (Supplementary Fig. 11). Furthermore, isoleucine deprived animals associated with *Ap* and *Lp* exhibited a decreased yeast appetite compared to the axenic animals. These data show that glucose and sucrose are interchangeable as carbon sources in what concerns the feeding behaviour of the flies.

**Egg laying assays**. After 72 h on HM with different AA compositions, groups of 16 females and 5 males were placed in apple juice agar plates (250 ml l⁻¹ apple juice, 19.5 g l⁻¹ agar, 20 g l⁻¹ sugar and 10 ml l⁻¹ nipagin [15% in ethanol]) and allowed to lay eggs for 24 h. Living adult flies and eggs were then counted and the number of eggs was normalised by the number of living females. Data were pooled from three experiments performed independently on different days.

**Assessment of bacterial growth in holidic media**. A liquid version of the HM was used to cultivate bacteria in vitro. The media was prepared by removing agar and cholesterol from the HM recipe to avoid turbidity in the media and replacing sucrose with 100 mM of glucose. An appropriate volume of cells was calculated to inoculate 20 ml of liquid HM with an initial optical density measured at 600 nm ($OD_{600}$) of 0.05 of each bacterium. The calculated volume of bacterial culture was centrifuged ($5000 \times g$, 10 min), washed once with 1x PBS and the pellet resuspended in 50 μl of 1x PBS. The cell suspension was used to inoculate 20 ml of liquid HM in 100 ml Erlenmeyer flasks, which were incubated at 25 °C with orbital agitation (180 rev min⁻¹). The number of CFU in the liquid cultures was assessed 0, 1, 2 and 3 days after the growth was resumed by plating and counting colonies as described above. The bacterial growth was determined for all tested conditions in at least three independent experiments.

**Fly head collection for metabolomics analysis**. For each measurement, at least 1000 mated gnotobiotic females maintained on each dietary condition were collected after brief CO₂ anaesthesia and snap-frozen in liquid nitrogen. To separate heads from the body and collect them, the frozen flies were vortexed in Eppendorf

tubes and sieved through a 710-mm and 425-mm mesh (Retsch GmbH). All the material used to handle the body parts was continuously cooled in liquid nitrogen throughout the process, to ensure that heads were kept frozen. At least 1000 heads per measurement were sent for metabolomics profiling as a paid service at Metabolon Inc, USA.

**Sample preparation for metabolomics analysis of fly heads**. Samples were prepared using the automated MicroLab STAR® system (Hamilton Company). Several recovery standards were added prior to the first step in the extraction process for quality control (QC) purposes. To remove protein, dissociate small molecules bound to protein or trapped in the precipitated protein matrix, and to recover chemically diverse metabolites, proteins were precipitated with methanol under vigorous shaking for 2 min (Glen Mills GenoGrinder 2000) followed by centrifugation. The resulting extract was divided into five fractions: two for analysis by two separate reverse phase (RP)/ultra-performance liquid chromatography (UPLC)-MS/MS methods with positive ion mode electrospray ionization (ESI), one for analysis by RP/UPLC-MS/MS with negative ion mode ESI, one for analysis by HILIC/UPLC-MS/MS with negative ion mode ESI, and one sample was reserved for backup. Samples were placed briefly on a TurboVap® (Zymark) to remove the organic solvent. The sample extracts were stored overnight under nitrogen before preparation for analysis. Samples were analysed in a Waters ACQUITY UPLC and a Thermo Scientific Q-Exactive high resolution/accurate mass spectrometer interfaced with a heated electrospray ionization (HESI-II) source and Orbitrap mass analyser operated at 35,000 mass resolution. The sample extract was dried then reconstituted in solvents compatible to each of the four methods. A pooled matrix sample generated by taking a small volume of each experimental sample served as a technical replicate throughout the data set. Each reconstitution solvent contained a series of standards at fixed concentrations to ensure injection and chromatographic consistency. Metabolites were identified by comparison to a library of authenticated standards that contains the retention time/index (RI), mass to charge ratio (*m/z*) and chromatographic data (including MS/MS spectral data) on all molecules present in the library. The plotted relative amount of metabolites detected in the analysis was normalised according to the number of heads in each sample.

**In silico analysis of the isoleucine biosynthetic pathway in *Ap***. The protein sequences were downloaded from NCBI (RefSeq Assembly accession ID: GCF_000193245.1) which is the contig level assembled genome of *Acetobacter pomorum* strain DM001. The amino acid sequences were aligned against the KEGG database (v_2014) using blastp with the parameters as max_target_seqs=10. The identified Blast output alignment was further parsed for identification of best hits and the hits were screened for KO IDs (enzymes) associated with isoleucine metabolism from BioCyc database. The identified pathway of isoleucine biosynthesis was mapped to the KEGG ID and used for pathway construction.

**Metabolomic analysis by LC–MS**. To trace the synthesis of the different metabolites produced by bacteria, two universally isotopically-labelled carbon sources were used: ¹³C₆-D-glucose (Cambridge Isotope Laboratories, #CLM-1396) or ¹³C₃-lactate (Cambridge Isotope Laboratories, #CLM-1579). In experiments using ¹³C₆-D-glucose, equimolar amounts of heavy glucose were used to completely substitute the glucose in the HM. In experiments using ¹³C₃-lactate, 20 g l⁻¹ of the heavy lactate were added to the HM formulation which normally does not contain lactate. A control culture with HM without isoleucine with unlabelled glucose was used in parallel. The liquid version of HM without cholesterol was used. To set up the cultures, the bacterial suspensions were centrifuged ($5000 \times g$, 10 min) and washed, and the pellet resuspended in 50 μl of 1x PBS. Five millilitres of HM were inoculated with an initial $OD_{600}$ of 0.05 in 25 ml Erlenmeyer flasks and incubated at 25 °C with orbital agitation (180 rev min⁻¹). A sample of the supernatant was collected after 24 h of incubation. For that, 50 μl of culture were centrifuged at $5000 \times g$ for 3 min at 4 °C and 10 μl of the supernatant immediately added to 500 μl of extraction buffer (30% methanol (Merck, #1.06035), 50% acetonitrile (Sigma-Aldrich, #900667) in Milli-Q water). Samples were kept at −80 °C until analysis. Before collecting the sample, the $OD_{600}$ of the cultures were measured to later normalise the values of the ¹³C-labelled metabolites by the growth rate of the bacteria. Liquid chromatography–mass spectrometry (LC–MS) was performed as described previously[94]: the supernatant samples were analysed on an LC–MS platform consisting of an Accela 600 LC system and an Exactive mass spectrometer (Thermo Scientific), using a ZIC-HILIC column (4.6 mm × 150 mm, 3.5 μm; Merck) with the mobile phase mixed by $A$ = water with 0.1% formic acid (v/v) and $B$ = acetonitrile with 0.1% formic acid. A gradient programme starting at 20% of $A$ and linearly increasing to 80% at 30 min was used followed by washing and re-equilibration steps. The total run time was 46 min. The LC stream was desolvated and ionised in the HESI probe. The Exactive mass spectrometer was operated in full-scan mode over a mass range of 75–1000 *m/z* at a resolution of 50,000 with polarity switching. LC–MS raw data was converted and analysed by LCquan (Thermo Scientific).

**Table 1 Parameters used for the pre-processing of raw metabolite peaks using xcms.**

| Parameters | Optimised values |
|---|---|
| Peak picking algorithm | Centwave |
| ppm | 25 |
| Min peak width | 6 |
| Max peak width | 60 |
| snthresh | 10 |
| Prefilter | [3,100] |
| mzdiff | −0.001 |
| Noise | 50,000 |

**LC-MS data pre-processing, correction, and peak identification.** Raw files were first pre-processed to extract the peaks and identify the metabolites. The pre-processing was performed using xcms[95]. The peaks were identified using the optimised parameters detailed in Table 1.

The spectra were aligned and corrected for retention time deviations using the obiwarp method with a bin size of 0.6. The peaks were further grouped across the samples using peak density for grouping with bin width = 5. The peak areas under the curve (AUC) were further corrected for the natural abundance of isotopes (for the isotopic tracer $^{13}C$) using the "Isocor" package[96]. The identification of metabolites was performed using an internal library of authentic chemical standards in the concentrations within the detectability range of the LC-MS column and conditions used to run samples.

To remove the levels of different metabolite isotopologues present in the HM ab initio, to highlight the changes in levels of metabolites and their isotopologues due to microbial metabolism, and to control for the variation caused by differences in bacterial growth rate among cultures, the peak areas were plotted after the following corrections:

The corrected peak areas (which in this case are the best estimates of the metabolite levels) of each of the isotopologues at 0-h growth (HM samples without bacteria, which represent the metabolite levels present ab initio in the medium) were subtracted from the peak areas of corresponding isotopologues measured in each of the experimental samples.

To scale for bacterial growth, peak areas were normalised using Eq. (1):

$$A_{i(Normalized)} = A_i * (5.15 / OD_{600,i}) \qquad (1)$$

Where $A_i$ = peak area of the metabolite in sample $i$ after correction for natural isotopes and $OD_{600,i}$ = $OD_{600}$ value of sample $i$ 5.15 is one of the highest $OD_{600}$ values measured in the 24 h $Ap/Lp$ co-cultures.

The metabolites were measured in three independent experiments. Metabolite levels before and after corrections are provided in Supplementary Data 1.

Mean percent enrichment (MPE) refers to the mean content of isotopic tracer in the measured metabolite. It was calculated using Eq. (2):

$$MPE = \left( \frac{\sum_{i=0}^{n} M_i \cdot i}{n} \right) \qquad (2)$$

Where

$M_i$ = relative proportion of isotopologue containing $i$ atoms.

$n$ = total number of atoms of the chemical element in the molecule.

**Graphical data representation and statistical analysis.** The statistical analysis and the graphical data representation shown in Figs. 1, 2, 4 and 6a, d and Supplementary Figs. 1–3, 10 and 11 were done using GraphPad Prism (version 6). The graphical data representation of Figs. 3 and 5a and Supplementary Figs. 4–9 were built in R using the ggplot2 package.

**Reporting summary.** Further information on research design is available in the Nature Research Reporting Summary linked to this article.

## Data availability

Source data are provided with this paper. Metabolite levels before and after corrections are provided in Supplementary Data 1. KEGG (v_2014) (https://www.genome.jp/kegg/kegg1.html) and BioCyc (https://biocyc.org/) databases were used to annotate the enzymes involved in isoleucine biosynthesis from the genome of *Acetobacter pomorum* strain. Source data are provided with this paper.

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

## Acknowledgements

We thank the laboratory of François Leulier (IGFL, France) for providing bacterial strains and Jessika Consuegra for advice on the proper cultivation conditions for bacterial strains and suggestions on how to test the effect of lactate on the host. We thank Alejandro Huerta Uribe for help with handling metabolomics data files. We thank Matthew Piper, Gili Ezra, Ibrahim Tastekin, Daniel Münch, and members of the Behaviour and Metabolism laboratory for helpful discussions and comments on the manuscript, and to Gil Costa for illustrations. We thank the Glass wash and media preparation and the fly platforms at the Champalimaud Centre for the Unknown for technical assistance. This project was supported by funding from the Kavli Foundation to C.R. The project leading to these results has received funding from "la Caixa" Banking Foundation to C.R. under the project code LCF/PR/HR17/52150002. A.P.F. is a member of the Champalimaud International Neuroscience Doctoral Programme and is supported by the FCT fellowship PD/BD/114277/2016. This project was supported by the Portuguese Foundation for Science and Technology (FCT) postdoctoral fellowship SFRH/BPD/79325/2011 to Z.C-S. Work by Z.C-S. was also financed by national funds through the FCT, in the framework of the financing of the Norma Transitória (DL 57/2016). O.D.K.M. and T.Z. were supported by the CRUK Career Development Fellowship, C53309/A19702. Research at the Centre for the Unknown is supported by the Champalimaud Foundation and the research infrastructure grant Congento LISBOA-01-0145-FEDER-022170.

## Author contributions

Conceptualization by S.F.H., and C.R.; Data curation and formal analysis by S.F.H., D.B.D., L.S., Z.C.-S., O.D.K.M. and C.R.; Investigation by S.F.H. and L.S. except from: metabolomics analysis by LC-MS which was conducted by T.Z. and acquisition of samples for metabolomics analysis in the heads of flies which was conducted by A.P.F., Z.C.-S., C. B., A.P.E. and M.A.; Supervision by C.R.; Validation by S.F.H., D.B.D. and C.R.; Visualisation by S.F.H, D.B.D., L.S. and C.R.; Writing – Original Draft by S.F.H. and C.R.; Writing – review and editing by S.F.H., D.B.D., L.S., Z.S., and C.R.; Project Administration by C.R.; Funding acquisition by C.R. All authors read and approved the final manuscript.

## Competing interests

The authors declare no competing interests.
