## [Peer Review File · Nature Communications]

Reviewers' comments:

Reviewer #1 (Remarks to the Author):

Silvia et al. utilized isotope tracing based metabolomics to discover a syntrophic relationship between *Acetobacter pomorum* (Ap) and *Lactobacillus plantarum* (Lp). In general, it is found that Lp provided lactate for Ap as a precursor to synthesize and secrete AAs. These AAs could allow Lp to overcome the imbalanced dairy. The story is potentially interesting. I was asked to carefully look at the metabolomics technical part. And I found there are severe technical issues with isotope tracing based metabolomics.

1. The authors did not mention in Methods whether they have corrected nature isotope abundances in the final isotopologue (not isotopomer) results. Some of the unlabeled results are not easy to explain. For example, in Figure 3 of ¹²C-glucose labeled samples, isoleucine has an observable M+4 peak but no M+3/M+5 peaks. If nature isotope abundances were corrected, M+1 to M+6 peaks of isoleucine should be close to 0. I would suspect the M+4 peak is a contamination of other metabolites. The authors should provide the raw MS peaks and details on this. The similar issues were also observed in Figure 5 and other figures.

2. For each metabolite, M+0 to M+n (n=6 for isoleucine) isotopologues should be demonstrated. But the authors only demonstrated M+3 to M+5 without explanations. I would like to see all isotopologue results with the correction of nature isotope abundances. It is common practice in isotope-tracing experiments.

3. The metabolic pathways from glucose or lactate to isoleucine are not explained or proposed. Since lactate has only 3 carbons, while isoleucine has 6 carbons. In general, in U-¹³C-lactate labeling experiment, it is reasonable that isoleucine has 3 carbons labelled from lactate. But in Figure 5, M+4 is the major isotopologue peak. How is lactate used for synthesize isoleucine (M+4)? It might be possible, but a metabolic pathway and explanation is required.

4. Similar question for U-¹³C-glucose labeling experiment, glucose has 6 carbons, but the authors only demonstrated isoleucine (M+5)? How is glucose used for synthesize isoleucine? Converted to lactate first? Then the same question repeated as comment 3.

5. In Figure 3, for the third and the fourth bars, the summed peak areas should be equal to each other (since they have the same experimental condition), but with different distributions on isotopologues. How to explain the differences on summed peak areas???
6. In addition, the MID pattern of isoleucine in figure 3 (second bar) and figure 5 (third bar) are quite different. How to explain the difference?
7. Page 11, line 269, “the LpWCFS1Δldh mutant strain failed to suppress yeast appetite (Fig. 4a)”. But actually, as the result showed, compared with the LpWCFS result, the LpWCFS1Δldh did not have significant changes for increased yeast appetite.
8. Figure 5, the authors should provide a scheme to show how M+3, M+4 and M+5 isoleucine were produced.
9. In the isotope-tracer experiments, could author use mole percentage enrichment (MPE) to quantify the new synthesized metabolites (isoleucine and other AAs)? It would help to clearly demonstrate the percentage of new synthesized metabolites.
10. It is very challenging to read supplementary Figure S4, please change to a color scheme. I will re-evaluate this figure in revision.
11. The details for identification of metabolite should be given.

Reviewer #2 (Remarks to the Author):

This well-written manuscript by Henriques et al examines uses a multiprong approach to understand how two bacteria, *Acetobacter Pomorum* (Ap) and *Lactobacillus plantarum* (Lp) metabolically interact within the context of *Drosophila melanogaster* host diet. By using an liquid chemically defined diet that mimics the *Drosophila* holidic diet, the authors examined how these two bacterial species grow in the absence of two essential amino acids – isoleucine and histidine. These experiments revealed that although Lp is an isoleucine auxotroph, coculture of Lp with Ap

allows Lp to grow in the absence of isoleucine. Using stable isotope tracers, the authors then demonstrate that in the presence of Lp, Ap synthesized and secreted labeled isoleucine. Moreover, the authors demonstrate the Lp synthesizes isoleucine using lactate provided by Ap, thus revealing an elegant symbiosis between the two bacteria. Finally, the authors provide evidence that this symbiotic relationship is likely not responsible for the ability of these two bacteria to jointly regulate *Drosophila* feeding behavior.

Overall, I found this a clear, logical, and important study. While the mechanism by which these two bacteria jointly regulate feeding behavior remains elusive, it doesn't diminish my interest in this story. In fact, I found it refreshing that the authors pointedly demonstrated that isoleucine production is essential for Lp growth but not fly behavior – a breath of fresh air in a field where these sort of metabolic interactions are usually the basis for wild speculation. While I can think of additional experiments that could be conducted by the authors, none would improve the manuscript.

Reviewer #3 (Remarks to the Author):

In this manuscript, Henriques et al. showed an intriguing metabolic connection between two dominant commensal bacterial species in *Drosophila* gut, *Acetobacter pomorum* (Ap) and *Lactobacillus plantarum* (Lp). Given their previous finding that these two gut microbes are required for controlling the appetite of the host, the authors attempted to identify the underlying mechanism. The careful gnotobiotic and metabolic experiments demonstrated the clear metabolic relay to produce essential amino acids from glucose via lactate. The in vitro data of bacterial culture using liquid holidic medium (prepared based on the fly diet) was convincing. Isotopic tracer experiments have beautifully demonstrated lactate is a product of glucose in Lp, and amino acids are produced from lactate in Ap. Lp is therefore required for the provision of lactate. The authors successfully showed that adding lactate is sufficient for Ap to influence the host behaviour. Amino acid production in Ap might also help Lp to grow under detrimental medium that lacks isoleucine. Overall, the manuscript is written well, addressing the essential question of "community-wise" bacterial metabolism and its effect on host physiology. However, it has still too many questions unanswered.

Major concerns:

1, Lack of identification of the factor regulating host appetite, as it is mentioned in the manuscript, is critical especially when the authors' take-home message seemed to be "host behaviour" altered by the microbial metabolic interaction shown in the title as well as in the abstract. What is only revealed in the study is that lactate mediates Lp's effect on the behaviour. This reviewer assumes that the rescue of increased appetite with Ile(-) diet by Ap/Lp would be because it promotes the growth of Ap. If it is the case, high abundance of Ap ($\sim 10^9$ CFU) but not Lp, during gnotobiotic preparation may rescue the behaviour.

2, Strikingly, the study did not reveal the mechanism of synergistic growth of the two bacterial species in vivo (fly gut/vial). Liquid culture nicely showed the unidirectional effect of Ap on Lp growth on Ile(-) medium. However, the authors did not demonstrate that this was because of the provision of Ile by Ap, especially given that the level of Ile production by Ap is very low. If possible, they need to use the loss of function mutant of Ap, which cannot utilise lactate or synthesise Ile. Besides, if lactate from Lp can also "feed" Ap, the growth of Ap in the liquid culture should be faster, as it was evident in the fly gut. Is it possible that lactate can increase Ap's growth in the fly gut, as well as in the liquid? The growth measurement in Fig.2 needs more time course, especially for the first 24 hours; one day of bacterial culture is too long to observe the upregulation of the growth speed. If this is not the case, then the bacterial growth in liquid is irrelevant to that in solid diet or in fly gut.

3, The major finding of the present study would be the production of Ile by Ap from Lp-derived lactate. However, this is expecting, given that Acetobacteraceae is proved to produce amino acids from lactate by using stable isotope labelling technique (Adler et al., *Appl Environ Microbiol* 80, 4702, 2014). Considering the study did not identify the Ap-derived factor controlling the host behaviour, the present may not significantly advance our understanding of microbial metabolism in the context of the host-microbe interaction. It should also be explained why Ap does not use glucose directly to produce Ile, while it produces (secretes) Ile by exogenous lactate.

Other concerns:

Fig.1a, It is possible that the quantity of bacteria, rather than quality, is important for the appetite control. It might be better to test the various amount of bacterial feeding on behaviour.

Fig.1c, Lp in Ile(-) is greater in amount ($\sim 10^5$) compared to that in complete medium (Lp alone conditions). What is the reason for this? There should be no growth at all under Ile (-) medium for this bacterial species, but the data suggests the opposite.

Fig.2b, Can the authors deny the possibility that the rescue of Lp growth by Ap is due to the provision of Ile by dead Ap?

Fig.3, How much did the total Ile content increased by co-culture?

Fig.3, Is the production of Ile abolished by co-culture of Ap with the Idh mutant Lp?

Champalimaud Foundation

We thank the reviewers for their constructive feedback. We are excited about the overall positive input and we have made a special effort to address all raised concerns, which has led to a substantially improved manuscript. Find below an overview of the major changes and a detailed point by point answer to the reviewer comments.

In this revised version of the manuscript we have incorporated all the feedback from reviewer #1. More specifically, we have reanalyzed all the metabolomics data using more stringent criteria, which included realigning the chromatogram peaks using a more stringent threshold for calculating the area under the peaks, and most importantly, have computationally corrected for the natural abundance of isotopes. This has led to substantially improved isotope resolved metabolomics data (Fig. 3 and 5a, and Supplementary Fig. 4 to 8). We now also include a pathway prediction for the biosynthesis of Isoleucine by *Acetobacter pomorum* which explains the measured isotopologue distributions (Fig. 5b). We would like to thank reviewer #1 for the constructive input, as implementing the suggestions has substantially improved our metabolomics data.

We have also made a substantial effort to experimentally address all points raised by reviewer #3. The revised manuscript includes a large set of new data which includes clear evidence that the effect of the commensal bacteria on behavior cannot be easily explained by an increase in *Ap* titres or microbial biomass and hence dietary proteins (Supplementary Fig. 2a and b). We now also show that bacterial biomass explains the impact of bacterial biomass on egg laying (Fig. 6c). This is a new finding, which explains how gut bacteria can modify a key life history trait. We now highlight this finding in the title and throughout the manuscript.

Reviewer #1 (Remarks to the Author):

Silivia et al. utilized isotope tracing based metabolomics to discover a syntrophic relationship between *Acetobacter pomorum* (*Ap*) and *Lactobacillus plantarum* (*Lp*). In general, it is found that *Lp* provided lactate for *Ap* as a precursor to synthesize and secrete

Champalimaud Foundation

AAs. These AAs could allow Lp to overcome the imbalanced diary. The story is potentially interesting. I was asked to carefully look at the metabolomics technical part. And I found there are severe technical issues with isotope tracing based metabolomics.

1. The authors did not mention in Methods whether they have corrected nature isotope abundances in the final isotopologue (not isotopomer) results. Some of the unlabeled results are not easy to explain. For example, in Figure 3 of ^{12}C -glucose labeled samples, isoleucine has an observable M+4 peak but no M+3/M+5 peaks. If nature isotope abundances were corrected, M+1 to M+6 peaks of isoleucine should be close to 0. I would suspect the M+4 peak is a contamination of other metabolites. The authors should provide the raw MS peaks and details on this. The similar issues were also observed in Figure 5 and other figures.

We systematically re-analyzed all the metabolomics data provided in the paper. This includes re-analysis of the peaks using more stringent computational approaches and the correction for natural isotope abundance in the final isotopologue amounts. More specifically we used more stringent parameters for peak extraction, removal of noise, retention time correction, and binning of the peaks (details are provided in the expanded Materials and Methods section lines 800 to 834). We now use the CentWave algorithm for peak picking with parameters optimized for our analysis (details can be found in the table in the Materials and Methods section line 804). This has increased the specificity of our analysis and has further removed contaminating peaks from co-eluting metabolites. Since leucine and Isoleucine are isomers having the same molecular weights and formula, earlier results were partially obscured due to contaminating peaks from leucine. Our more rigorous approach has now addressed this issue to a large extent. Especially, because we use very conservative settings to ensure that what we plot is indeed present while risking to miss relevant data. Importantly this reanalysis does not change any of our conclusions. As requested, we also provide the raw MS peaks with this submission (*Henriques Raw Spectral Data for Reviewer Positive Mode* and *Henriques Raw Spectral Data for Reviewer Negative Mode*).

Champalimaud Foundation

The reanalysis which incorporates the input of reviewer #1 has led to much cleaner data. This is evidenced by there not being a detected signal for Isoleucine isotopologues in most controls or the detected labelling pattern being in agreement with the most likely biosynthesis steps for Isoleucine by *Ap*, which we have reconstructed from genome information and now provide in Fig. 5b.

The reanalyzed data and the pathway prediction now also addresses the issues concerning the labelling pattern mentioned by the reviewer. The problematic labelling patterns are either not present anymore (and were hence most likely due to the natural isotope contribution) or explained by the biosynthesis pathway. In short, given that only two labelled carbon atoms are provided by lactate, one expects to mainly see the m+2 form in the Isoleucine synthesized by *Ap* (Fig. 5b). And this is exactly what we see in our data (Fig. 3 and 5a). We have also significantly expanded the Materials and Methods section to describe in greater details the methods used for the isotope resolved metabolomics data in the manuscript (lines 800 to 834).

Please note that we removed one metabolomics dataset. Due to the use of the more stringent criteria for the identification of metabolites, in some of the measurements in the 24 h culture of *Acetobacter pomorum* with ¹³C-labeled lactate we were not able to detect significant amounts of amino acids. This is likely due to these samples being the ones in which bacteria grew less. As the 24 h and 48 h samples are to a large extent redundant we therefore decided to exclude the resulting noisy 24 h dataset.

2. For each metabolite, M+0 to M+n (n=6 for isoleucine) isotopologues should be demonstrated. But the authors only demonstrated M+3 to M+5 without explanations. I would like to see all isotopologue results with the correction of nature isotope abundances. It is common practice in isotope-tracing experiments.

In the original version of the manuscript we had opted not to show the m+0 form as in the experiments in which Isoleucine is not removed from the diet, the m+0 signal eclipsed the signal from all other forms. To solve this problem we now plot the

Champalimaud Foundation

changes in the m+0 form over time as a separate plot (Supplementary Fig. 5 and 8). Since our aim is to show the production/consumption of metabolites from labelled glucose/lactate we have separately provided plots for the production of higher isotopologues (m+n; n>0) and the m+0 form of metabolites. In the figures in which we plot the higher isotopologues (m+1 to m+6) we now include all isotopologue forms (Fig. 3 and 5a, and Supplementary Fig. 4 and 7). To be as transparent as possible, we now provide a table containing the raw area under the curve values, the values after correction for natural isotope abundance, and the normalized values after correction for OD for each isotopologues in all samples for the metabolites analyzed in the manuscript (Supplementary Table 2).

3. The metabolic pathways from glucose or lactate to isoleucine are not explained or proposed. Since lactate has only 3 carbons, while isoleucine has 6 carbons. In general, in U-13C-lactate labeling experiment, it is reasonable that isoleucine has 3 carbons labelled from lactate. But in Figure 5, M+4 is the major isotopologue peak. How is lactate used for synthesise isoleucine (M+4)? It might be possible, but a metabolic pathway and explanation is required.

4. Similar question for U-13C-glucose labeling experiment, glucose has 6 carbons, but the authors only demonstrated isoleucine (M+5)? How is glucose used for synthesise isoleucine? Converted to lactate first? Then the same question repeated as comment 3.

We now include a predicted metabolic pathway for the synthesis of Isoleucine by *Ap* (Fig. 5b) and discuss it in a new section of the paper (lines 351 to 375). We base our pathway prediction on the literature and on genome information to verify the presence of all relevant enzymes in the *Ap* genome (Supplementary Table 1). The derived pathway proposes that lactate can either be fed into the TCA cycle after conversion into pyruvate to be transformed into Oxaloacetate, Aspartate, and finally Isoleucine. Or that it can be directly fed into the Aspartate-derived biosynthetic pathway of Isoleucine. A key prediction is that lactate metabolized through this

Champalimaud Foundation

pathway only provides two carbons at a time to Isoleucine biosynthesis leading to the enrichment in the m+2 form of this AA. This is exactly what we observe in the reanalyzed cleaned up metabolomics data provided in our revised manuscript (Fig. 3 and 5a).

Our model also predicts that Aspartate is a key precursor for the synthesis of Isoleucine. This prediction is backed up by the observation that concomitantly with an increase in Isoleucine biosynthesis we observe a decrease in the concentration of Aspartate after 48 hours (Supplementary Fig. 5 and 8 and Supplementary Table 2). This strongly suggests that when possible *Ap* also uses Aspartate from the media for the synthesis of Isoleucine.

We therefore now provide a genome-based prediction of the pathway for Isoleucine biosynthesis from lactate by *Ap*, which is in agreement with the labelling data and other changes in our metabolomics measurements. This makes us confident that the provided model is correct. We feel that the addition of the metabolic model for Isoleucine biosynthesis is an important new addition to our manuscript and significantly enhances its value.

5. In Figure 3, for the third and the fourth bars, the summed peak areas should be equal to each other (since they have the same experimental condition), but with different distributions on isotopologues. How to explain the differences on summed peak areas???

One would indeed have expected that the total peak areas in the plot we showed in the original version of our manuscript should be equivalent under the same conditions. However, for the third row (the ^{12}C condition) one would have expected that the overwhelming isotopologue should be the m+0 form. As the m+0 form was not shown in the plot this explains why the summed bar height was so different.

Champalimaud Foundation

6. In addition, the MID pattern of isoleucine in figure 3 (second bar) and figure 5 (third bar) are quite different. How to explain the difference?

In the now provided revised data, which was reanalyzed according to the suggestions of reviewer #1, the MID patterns of Isoleucine production from glucose (Fig. 3) and lactate (Fig. 5a) are very similar in their distribution. The m+2 form is clearly the most abundant isotopologue in both experiments. All provided MID patterns are therefore now coherent with the prediction that glucose is transformed into lactate which is then used by *Ap* to synthesize Isoleucine.

7. Page 11, line 269, "the *Lp*^{WCFS1Δldh} mutant strain failed to suppress yeast appetite (Fig. 4a)". But actually, as the result showed, compared with the *Lp*^{WCFS} result, the *Lp*^{WCFS1Δldh} did not have significant changes for increased yeast appetite.

We apologize if we have not been clear in the display of our data. As is evident from the plot in Fig. 4a, that there is a strong difference between the *Lp*^{WCFS1} and the *Lp*^{WCFS1Δldh} condition. Although we do not show it, the difference is strongly statistically significant. For reasons of simplicity we decided to only show the outcome of the statistical test for the comparison to the germ-free condition. That is what the ns indicates for the *Lp*^{WCFS1Δldh} condition. While the *Lp*^{WCFS1} strongly suppresses yeast feeding as shown by the strongly statistically significant difference when compared to the germ-free condition, the *Lp*^{WCFS1Δldh} condition does not change yeast feeding when compared to the germ-free condition.

8. Figure 5, the authors should provide a scheme to show how M+3, M+4 and M+5 isoleucine were produced.

Champalimaud Foundation

In Fig. 5b we now provide the likely metabolic pathway by which *Ap* produces Isoleucine.

9. In the isotope-tracer experiments, could author use mole percentage enrichment (MPE) to quantify the new synthesized metabolites (isoleucine and other AAs)? It would help to clearly demonstrate the percentage of new synthesized metabolites.

We thank the reviewer for this excellent suggestion. We now include the MPE values in the text (lines 263 to 266 and 334 to 336) as well as Supplementary Table 2.

10. It is very challenging to read supplementary Figure S4, please change to a color scheme. I will re-evaluate this figure in revision.

We apologize if the used color scheme was not helpful in analyzing the data. We have now used ColorBrewer to generate a new color palette which is both compatible with color blindness, optimizes the visual separation of the different isotopologues, and is esthetically pleasing. To do so we had however to sacrifice the unified color scheme for the different dietary conditions which we had used in the original version of the manuscript. We hope that this helps with the assessment of our data.

11. The details for identification of metabolite should be given.

We now provide a detailed account of how the metabolites were identified in the Materials and Methods section (lines 800 to 812). In short, metabolites were identified using an internal library of authentic chemical standards run on the HILIC column using similar conditions as the ones used in our experiments.

Champalimaud Foundation

Reviewer #2 (Remarks to the Author):

This well-written manuscript by Henriques et al. examines uses a multiprong approach to understand how two bacteria, *Acetobacter Pomorum* (Ap) and *Lactobacillus plantarum* (Lp) metabolically interact within the context of *Drosophila melanogaster* host diet. By using an liquid chemically defined diet that mimics the *Drosophila* holidic diet, the authors examined how these two bacterial species grow in the absence of two essential amino acids - isoleucine and histidine. These experiments revealed that although Lp is an isoleucine auxotroph, coculture of Lp with Ap allows Lp to grow in the absence of isoleucine. Using stable isotope tracers, the authors then demonstrate that in the presence of Lp, Ap synthesized and secreted labeled isoleucine. Moreover, the authors demonstrate the Lp synthesizes isoleucine using lactate provided by Ap, thus revealing an elegant symbiosis between the two bacteria. Finally, the authors provide evidence that this symbiotic relationship is likely not responsible for the ability of these two bacteria to jointly regulate *Drosophila* feeding behavior.

Overall, I found this a clear, logical, and important study. While the mechanism by which these two bacteria jointly regulate feeding behavior remains elusive, it doesn't diminish my interest in this story. In fact, I found it refreshing that the authors pointedly demonstrated that isoleucine production is essential for Lp growth but not fly behavior - a breath of fresh air in a field where these sort of metabolic interactions are usually the basis for wild speculation. While I can think of additional experiments that could be conducted by the authors, none would improve the manuscript.

We thank the reviewer for the enthusiastic assessment of our work.

Reviewer #3 (Remarks to the Author):

In this manuscript, Henriques et al. showed an intriguing metabolic connection between two dominant commensal bacterial species in *Drosophila* gut, *Acetobacter pomorum* (Ap) and *Lactobacillus plantarum* (Lp). Given their previous finding that these two gut microbes are required for controlling the appetite of the host, the authors attempted to identify the underlying mechanism. The careful gnotobiotic and metabolic experiments demonstrated the clear metabolic relay to produce essential amino acids from glucose via lactate. The in vitro data of bacterial culture using liquid holidic

Champalimaud Foundation

medium (prepared based on the fly diet) was convincing. Isotopic tracer experiments have beautifully demonstrated lactate is a product of glucose in Lp, and amino acids are produced from lactate in Ap. Lp is therefore required for the provision of lactate. The authors successfully showed that adding lactate is sufficient for Ap to influence the host behaviour. Amino acid production in Ap might also help Lp to grow under detrimental medium that lacks isoleucine. Overall, the manuscript is written well, addressing the essential question of "community-wise" bacterial metabolism and its effect on host physiology. However, it has still too many questions unanswered.

We thank the reviewer for the positive assessment of our work. We have made a significant effort to answer the questions raised by the reviewer by performing additional experiments which in our opinion address most of the raised concerns.

Major concerns:

1, Lack of identification of the factor regulating host appetite, as it is mentioned in the manuscript, is critical especially when the authors' take-home message seemed to be "host behaviour" altered by the microbial metabolic interaction shown in the title as well as in the abstract. What is only revealed in the study is that lactate mediates Lp's effect on the behaviour.

While we would have loved to identify the molecules produced by the commensal bacteria that alter behavior we never claimed that we aimed at identifying them in this work. Our specific goal was to identify why two bacteria species are required to alter protein appetite in the fly. While the identified mechanism allowing the microbial community to overcome diets without essential amino acids does not seem to account for the mechanism by which the commensal bacteria alter behavior, they are still essential for their ability to do so. If the bacterial community would not establish a syntrophic relationship they would not survive in the host when exposed to a diet deficient in essential amino acids. As such this mechanism is essential to explain how the bacteria can alter host behavior.

Champalimaud Foundation

We would also like to highlight that thanks to experiments suggested by the reviewer we were able to show that commensal bacteria increase egg laying by providing biomass (i.e. amino acids) (Fig. 6c – replaces the previous result in which the *Ap* + lactate condition did not show changes in egg laying). As such the ability of the syntrophic community to sustain the production of limiting and essential amino acids is a key process by which egg production is increased in malnourished animals with a microbiome.

This reviewer assumes that the rescue of increased appetite with Ile(-) diet by *Ap/Lp* would be because it promotes the growth of *Ap*. If it is the case, high abundance of *Ap* ($\sim 10^9$ CFU) but not *Lp*, during gnotobiotic preparation may rescue the behaviour.

We agree with the reviewer that one of the most straight forward interpretations of the bacterial growth data in the host would be that the increase in *Ap* titres observed when this bacterium is co-cultivated with *Lp* could be sufficient to alter protein appetite. We however now provide data which suggest that this is not the case. First of all, when we associate flies with a high abundance of *Ap* alone (10^9 as suggested by the reviewer) we observe no change in protein appetite when compared to germ-free controls (Supplementary Fig. 2a). This observation is backed up by an additional experiment in which the supplementation of the fly culture with high titres of heat-killed *Ap* and *Lp* did not lead to a reduction in protein appetite (Supplementary Fig. 2b). Please note, that we added 1.01×10^9 CFU of *Ap* and 1.31×10^9 CFU of *Lp* on three consecutive days before the behavioral experiments were performed. This corresponds to the bacterial titres measured in the food of the flies after three days of growth. This is a very conservative experimental design, given that in the experiment with living bacteria, before the exponential growth starts, the bacterial titres are much lower. We are therefore adding a clear overabundance of biomass and still see no evidence for a reduction in protein appetite. Together with results published earlier in Leitão-Gonçalves *et al.*, PLoS Biology, 2018, this makes it

Champalimaud Foundation

very unlikely that the effect of *Lp* on behavior is limited to boosting the titres of *Ap* and the increase in biomass available to the host. We now show however, that this mechanism is likely to be sufficient to improve egg laying in the host.

2, Strikingly, the study did not reveal the mechanism of synergistic growth of the two bacterial species in vivo (fly gut/vial). Liquid culture nicely showed the unidirectional effect of *Ap* on *Lp* growth on Ile(-) medium. However, the authors did not demonstrate that this was because of the provision of Ile by *Ap*, especially given that the level of Ile production by *Ap* is very low. If possible, they need to use the loss of function mutant of *Ap*, which cannot utilise lactate or synthesise Ile.

We thank the reviewer for the suggestion to use bacterial genetics to complement our metabolomics data. Unfortunately the genetic tools to perform targeted genome engineering in *Acetobacteraceae* are very limited. It is therefore very difficult to engineer mutant *Ap* for specific genes. Given this limitation, we decided to test mutant *Acetobacter fabarum* (*Af*) strains from a mutant library generated by the Chaston laboratory (White *et al.*, G3, 2018).

We identified multiple strains in the library with putative transposon insertions in genes encoding subunits of lactate dehydrogenase, sequence-verified the insertions, and tested the strains for a lack in their ability to utilize lactate as a carbon source. We did so by growing the strains in a minimal medium containing lactate as the sole carbon source and urea as the nitrogen source. Surprisingly the mutant strains all grew in this medium suggesting that they retained the ability to utilize lactate. After careful analysis of *Af* genome we found multiple enzymes that could mediate lactate utilization in this bacterium. These observations could easily explain why mutant strains devoid of single lactate dehydrogenase subunits can still utilize this carbon source. In this scenario it would be necessary to generate double or triple mutants strain to abolish the ability of *Af* to metabolize lactate. This would be a major undertaking that would have been hardly feasible during the available time and

Champalimaud Foundation

became impossible due to the current lockdown due to the COVID crisis. Importantly, given the strong data provided in the current version of the manuscript we feel that our current conclusions are supported convincingly even without the inclusion of the requested bacterial mutant data.

We also considered testing *Af* strains with putative mutations in genes encoding enzymes required for the biosynthesis of Isoleucine. But we rapidly dropped this plan as such mutants would very likely not grow in a medium lacking Isoleucine. This would make any behavioral and bacterial growth result meaningless.

It is important to reiterate that we think that our conclusion that *Ap* provides Isoleucine for *Lp* to grow in media lacking this AA is valid despite our inability to test that genetically (albeit as a geneticist I agree that this would be elegant). It is clear that *Lp* cannot grow without Isoleucine. It is also clear that *Ap* rescues this metabolic deficiency and given the data we have until this point, including the inability of *Lp* to synthesize Isoleucine and isotope resolved metabolomics, the only possible explanation for this phenomenon is that *Ap* provides *Lp* with this AA.

Besides, if lactate from *Lp* can also "feed" *Ap*, the growth of *Ap* in the liquid culture should be faster, as it was evident in the fly gut. Is it possible that lactate can increase *Ap*'s growth in the fly gut, as well as in the liquid? The growth measurement in Fig.2 needs more time course, especially for the first 24 hours; one day of bacterial culture is too long to observe the upregulation of the growth speed. If this is not the case, then the bacterial growth in liquid is irrelevant to that in solid diet or in fly gut.

We thank the reviewer for the suggestion to strengthen the liquid culture growth data by adding more time points. We have now done so and now include these data in the edited version of Fig. 2. We see no clear evidence for *Ap* to be able to grow more efficiently in the presence of either *Lp* or lactate in the liquid culture. A very careful dissection of a possible accelerated growth of *Ap* is however made difficult by the

Champalimaud Foundation

decrease in bacterial titres in the first hours of the liquid culture. This is very likely to be caused by the shift in medium when we introduce the bacteria to the liquid holidic diet. While this can be avoided by pre-inoculating the bacteria in holidic medium, we preferred to mimic the dietary conditions the bacteria will undergo in the *in vivo* host experiments and have therefore restrained from doing so.

It is important to highlight (as we do in the paper) that our goal was not to fully mimic the conditions in the fly but to generate a model where we can isolate the effect of the dietary content on the bacteria. While this model will not recapitulate all dietary effects, especially if they rely on an interaction with the host, it is still able to reveal important dietary vulnerabilities of the microbes, like the inability to grow in diets lacking specific AAs. As such we could not only identify the ability of *Ap* to support *Lp* growth in media lacking Ile, but we were also, for example, able to observe that both in liquid co-cultures and in the fly, *Lp* grew to higher titres in complete media compared to media lacking Isoleucine (Fig. 2b and Fig. 1c, respectively). Furthermore, the *Ap* bacteria in co-culture grew to identical titres in both dietary treatments, which was also observed in the host (Fig. 1b vs Fig. 2a). The *in vitro* culture condition therefore recapitulates many dietary effects observed in the fly.

It is however clear that the conditions in the fly gut will be much harsher than in the *in vitro* culture system and that there will always be a limit to what can be recapitulated in the *in vitro* system. Bacteria will for instance be killed in the host digestive system at a very high rate (see for example Storelli *et al.*, Cell metabolism, 2018). They might also be subjected to other, novel metabolic challenges. It is very likely that in such deleterious situations a growth advantage will reveal itself as a clear difference in the *Ap* bacterial titres inside the host, something we do not observe for *Ap* in the liquid culture. This straight forward difference could explain the difference between the growth rate observed *in vitro* and in the host pointed out by the reviewer.

3, The major finding of the present study would be the production of Ile by *Ap* from *Lp*-derived lactate. However, this is expecting, given

Champalimaud Foundation

that Acetobacteraceae is proved to produce amino acids from lactate by using stable isotope labelling technique (Adler et al., Appl Environ, Microbiol 80, 4702, 2014). Considering the study did not identify the *Ap*-derived factor controlling the host behaviour, the present may not significantly advance our understanding of microbial metabolism in the context of the host-microbe interaction. It should also be explained why *Ap* does not use glucose directly to produce Ile, while it produces (secretes) Ile by exogenous lactate.

We now provide a detailed prediction of the metabolic pathway for the biosynthesis of Isoleucine by *Ap* from lactate (Fig. 5b). This pathway is based on genome information and is backed up by the metabolomics data. Importantly, we do not see a significant production of m+0 forms of Isoleucine in the experiments in which we use ¹³C labelled lactate (Fig. 5a). This means that under these conditions the contribution of metabolites containing ¹²C carbons is neglectable. When lactate is present glucose is therefore not contributing to the synthesis of Isoleucine. Please note that given that *Ap* grows in medium lacking Isoleucine and lactate (Fig. 2a) *Ap* must be able to synthesize Isoleucine in the absence of lactate. When present lactate however is clearly the preferred substrate for Isoleucine synthesis, allowing *Ap* to produce much higher amounts of this amino acid. Why this is the case remains unclear.

Other concerns:

Fig.1a, It is possible that the quantity of bacteria, rather than quality, is important for the appetite control. It might be better to test the various amount of bacterial feeding on behaviour.

As discussed above we are quite certain that the effect of how the co-culture alters behavior cannot be explained simply by the increase in bacterial titre. In Leitão-Gonçalves *et al.*, PLoS Biology, 2017 we had shown that commensal bacteria are able to alter behavior even if their titre is reduced and that associating flies with heat-killed bacteria does not lead to a change in behavior. We now also show that associating flies with a high titre of *Ap* does not alter yeast feeding (Supplementary

Champalimaud Foundation

Fig. 2a) and that adding exaggeratedly high amounts of heat-killed bacteria does also not alter behavior (Supplementary Fig. 2b). Taken together these data suggest that the increase in bacterial titre or bacterial biomass is not sufficient to explain the mechanism by which the co-culture alters behavior.

Fig.1c, *Lp* in Ile(-) is greater in amount ($\sim 10^5$) compared to that in complete medium (*Lp* alone conditions). What is the reason for this? There should be no growth at all under Ile (-) medium for this bacterial species, but the data suggests the opposite.

We apologize if the way we plotted the data is confusing. The *Lp* titres in the – Isoleucine situation are not higher than in the complete medium condition. This impression must have been caused by the difference in the scale in which we plotted the *Lp* titres. We changed the scales in the plots in which we report the bacterial titres and hope that the data are now easier to read (Fig. 1b and c, and Supplementary Fig.1b and c).

Fig.2b, Can the authors deny the possibility that the rescue of *Lp* growth by *Ap* is due to the provision of Ile by dead *Ap*?

As we have not explicitly tested this we can neither support nor deny this possibility. It is important to note that we do not make any statement as to how *Ap* provides *Lp* with amino acids. What we know from our metabolomics measurements is that in medium lacking Isoleucine and when in culture with *Lp* or lactate, *Ap* secretes *de novo* synthesized amino acids. This does in theory not preclude amino acids from dead *Ap* to be available for *Lp* growth.

Champalimaud Foundation

Fig.3, How much did the total Ile content increased by co-culture?

The total Isoleucine levels (adding up all isotopologues) produced and secreted in media lacking Isoleucine are as follows:

Culture	Medium	Time point	Mean Peak AUC
Ap (monoculture)	holidic medium -Ile	24 hour	0
Ap/Lp (co-culture)	holidic medium -Ile	24 hour	17805372.56
Ap (monoculture)	holidic medium -Ile	48 hour	3576973
Ap/Lp (co-culture)	holidic medium -Ile	48 hour	182637594

As is easily seen we cannot detect Isoleucine after 24 hours in the *Ap* monoculture while the co-culture condition produced and secreted significant amounts of this amino acid. Likewise, after 48 hours in the co-culture condition the Isoleucine levels in the medium are 51 fold higher in *Ap/Lp* co-culture compared to monoculture of *Ap*.

Fig.3, Is the production of Ile abolished by co-culture of *Ap* with the *ldh* mutant *Lp*?

Unfortunately, we did not perform metabolomics in *Lp*^{WCFS1 Δ ldh} co-cultures and can therefore not answer this question.

REVIEWERS' COMMENTS:

Reviewer #1 (Remarks to the Author):

In the revised manuscript, metabolomics data quality has been significantly improved. Major questions were fully addressed by the authors. Congratulations!

I only have two minor comments:

1. For original comment 1, please plot the raw MS peaks for the metabolite isoleucine, and provide the data as a supplementary figure.
2. In Figure 5, the authors provided the predicted metabolic pathway. Could they explain how the isotopologue M+5 and M+6 were generated?

Reviewer #3 (Remarks to the Author):

The authors have made a tremendous effort, despite the difficult situation of the world, to answer this reviewer's concerns by performing necessary experiments. Now that the conclusion of the study is supported by the data presented in the revised manuscript, it would be suitable for publication.

Champalimaud Foundation

Reviewer #1 (Remarks to the Author):

In the revised manuscript, metabolomics data quality has been significantly improved. Major questions were fully addressed by the authors. Congratulations!

We thank the reviewer for the kind words.

I only have two minor comments:

1. For original comment 1, please plot the raw MS peaks for the metabolite isoleucine, and provide the data as a supplementary figure.

We thank the reviewer for this excellent suggestion. We now plot the raw MS peak for the metabolite Isoleucine in positive ion mode at the specific RT of 730 seconds as Supplementary Figure 4. In panel a) we show the distribution of ions for all the Isoleucine isotopologues (m+0 to m+6) of a single *Ap* culture and in panel b) for a sample of an *Ap/Lp* co-culture. We also refer to this figure in line 253 of the manuscript:

“Clear differences in the pattern of isotopologue distributions between the *Ap* alone and the co-culture condition were already clearly visible at the level of the raw MS peaks of Isoleucine (Supplementary Figure 4). ”

2. In Figure 5, the authors provided the predicted metabolic pathway. Could they explain how the isotopologue M+5 and M+6 were generated?

This is an excellent question. One possible explanation would be that oxaloacetate produced from the labelled lactate can be labelled at the CO-COOH or CH₂-COOH positions as shown in Fig. 5b. The TCA cycle leads to the buildup of labelled Citrate which then undergoes decarboxylation steps in the TCA cycle leading to the formation of uneven isotopologues of oxaloacetate. Given that the TCA cycle is

Champalimaud Foundation

recurring, the emergence of heavier isotopologues of oxaloacetate will be eventually observed after multiple cycles. This would then lead to the generation of heavier isotopologues of aspartate which would contribute to generating heavier ($m+5$ and $m+6$) isotopologues of Isoleucine.